

# Dynamical localization and slow thermalization in a class of disorder-free periodically driven one-dimensional interacting systems

Sreemayee Aditya[1⋆] and Diptiman Sen[1,2†]

**1** Center for High Energy Physics, Indian Institute of Science, Bengaluru 560012, India
**2** Department of Physics, Indian Institute of Science, Bengaluru 560012, India

⋆ sreemayeea@iisc.ac.in , † diptiman@iisc.ac.in

## Abstract

We study if the interplay between dynamical localization and interactions in periodically driven quantum systems can give rise to anomalous thermalization behavior. Specifically, we consider one-dimensional models with interacting spinless fermions with nearest-neighbor hopping and density-density interactions, and a periodically driven on-site potential with spatial periodicity $m = 2$ and $m = 4$. At a dynamical localization point, these models evade thermalization either due to the presence of an extensive number of conserved quantities (for weak interactions) or due to the kinetic constraints caused by drive-induced resonances (for strong interactions). Our models therefore illustrate interesting mechanisms for generating constrained dynamics in Floquet systems which are difficult to realize in an undriven system.

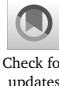

# 1 Introduction

The non-equilibrium dynamics of quantum systems has been extensively studied in recent years [1–13]. Various kinds of time-dependent protocols have been considered such as quenching and ramping [1–4, 14–29], periodic driving [5–13], and quasiperiodic and aperiodic driving [30–38]. There have been several experimental studies of non-equilibrium dynamics in systems of cold atoms trapped in optical lattices [39–48].

Periodic driving of quantum systems can give rise to a host of interesting phenomena which have no equilibrium counterparts, such as the generation of drive-induced topological phases [49–54], Floquet time crystals [55–57], dynamical localization [58–64], dynamical freezing [65–70], tuning between ergodic and non-ergodic behaviors [71–73], and dynamical transitions [74–79]. The out-of-equilibrium dynamics of a wide class of closes quantum systems is believed to be governed by the eigenstate thermalization hypothesis (ETH) [80–84]. According to ETH, all the eigenstates near the middle of the energy spectrum of a closed, non-integrable and disorder-free quantum system are thermal; the thermal nature of such states guarantees the ergodicity of the system. However, some instances are known where ETH is violated, for example, in integrable quantum systems and in many-body localized phases in one dimension in the presence of disorder and interactions [85, 86]. In recent years, it has been found that ETH can be broken in some quantum systems which are not integrable and have no disorder. The breaking of ETH may be weak or strong. In the case of weak ergodicity breaking, systems evade ergodicity due to the presence of quantum many-body scars [87–98]. Quantum many-body scars are states which lie near the middle of the spectrum and have anomalously low entanglement entropy between two halves of the system. The number of scar states is typically much smaller than the full Hilbert space dimension. Moreover, the scar states form a subspace which is almost decoupled from the thermal subspace. Hence they are protected from thermalization for a long period of time and show persistent long-time coherent oscillations in their dynamics; this has been observed recently in Rydberg atoms simulators. Furthermore, the interplay between quantum many-body scar states and periodic driving can generate rich dynamical phase diagrams which have been studied in a number of papers [73, 99–104]. One possible mechanism for strong ergodicity breaking is Hilbert space fragmentation (HSF) [105–109] which occurs due to the presence of certain kinetic constraints in the dynamics. These kinetic constraints lead to the fragmentation of the Hilbert space into many disconnected sectors which can give rise to non-ergodic behavior in such systems. The HSF in Floquet systems has been examined recently [110].

It has been shown in a series of theoretical works that quantum many-body scar states can appear in systems hosting flat bands supported by compact localization [94–96]. Motivated by this idea, we will pose a similar question in the context of Floquet systems undergoing dynamical localization (DL) [58–64]. DL can be achieved in Floquet systems by tuning some

of the system parameters which makes the effective hopping amplitudes zero or very small. DL can thus be a powerful tool for generating flat bands. The effects of interactions then become predominant which makes such systems highly promising platforms for investigating correlated out-of-equilibrium phases of matter. DL is special to periodically driven systems since this phenomenon has no equilibrium analog. Furthermore, the interplay between DL and quasiperiodic driving can prevent thermalization in a quantum system; this has been studied recently [111]. Having understood the rich possibilities that DL can offer in Floquet systems, the natural question that motivates our work is as follows. Can the interplay between DL and interactions in a one-dimensional disorder-free periodically driven closed quantum system induce anomalous thermalization behavior?

Table 1: Schematic of main results obtained for the period-2 and period-4 models. $\mu$ and $\omega$ denote the driving amplitude and frequency respectively.

| Class of periodic potential | Condition for dynamical localization | Dynamical localization and $\mu \gg J, V$ | Dynamical localization and $\mu = V \gg J$ |
|---|---|---|---|
| $m = 2$ | $\mu = n\omega \, (n = 1, 2, \cdots)$ | cMany-body flat bands, slow thermalization due to an emergent integrability | Model of Hilbert space fragmentation |
| $m = 4 \, (\phi = 0)$ | $\mu = 2n\omega \, (n = 1, 2, \cdots)$ | Same as period-2 model | Same as the period-2 model |
| $m = 4 \, (\phi = 7\pi/4)$ | $\mu = n\omega \, (n = 1, 2, \cdots)$ | Many low-entanglement states near the middle of the spectrum due to the presence of an extensive number of conserved quantities | Model of Hilbert space fragmentation but different from the period-2 case |

Another motivation for our work is as follows. As mentioned above, it has been known for many years that periodic driving of non-interacting systems can be used to generate quantum systems with a wide variety of band structures. It is therefore natural to ask if periodic driving of interacting systems can produce new kinds of quantum many-body systems whose parameters can be readily tuned.

The plan of this paper is as follows. In Sec. 2, we introduce our general model which consists of a one-dimensional system of spinless fermions with nearest-neighbor hopping, an on-site potential which is periodic in space [112] and is also driven periodically in time, and a density-density interaction between nearest-neighbor sites. In Sec. 3, we study in detail a model in which the potential has a periodicity of 2 sites. We first use first-order Floquet perturbation theory to derive an effective Floquet Hamiltonian for a non-interacting system when the driving amplitude and frequency are much larger than the hopping amplitude. We find that the system shows DL for certain values of the system parameters. We look at a two-point correlation function as a function of the stroboscopic time $t = nT$, where $T = 2\pi/\omega$ is the time period. We find that the correlator can decay as a power-law, where the power depends on the structure of the quasienergy dispersion around zero momentum. An interesting dynamical phase transition is found to occur when the dispersion changes, and a crossover between different powers occurs. Next, we look at the effects of interactions on DL. Exactly

at a DL point, we find that there is an emergent integrability with a large number of conserved quantities; these quantities are just the fermion occupation numbers (0 or 1) at the different sites. An exact numerical calculation of the time evolution of the system confirms this result from the Floquet Hamiltonian, namely, we find that the two-point correlation function and Loschmidt echo (which is the overlap between an initial state and its time-evolved state) oscillate in time and show almost perfect revivals with a frequency which depends on the interaction strength. The integrability disappears as we go away from a DL point, and the Loschmidt echo then decays rapidly with time. In Sec. 3.4, we consider the combined effects of resonances [113] and DL for the period-2 model. The first-order Floquet Hamiltonian then has a remarkable structure in which we have both the large number of conserved quantities as well as a density-dependent hopping in which the hopping between sites $j + 1$ and $j + 2$ depends on the occupation numbers on their neighboring sites $j$ and $j + 3$. We thus obtain dynamical constraints on the hopping. This leads to the appearance of an exponentially large number of zero quasienergy states (an expression for this number is derived in Appendix A using a transfer matrix method), a highly fragmented Hilbert space, and several states with low entanglement entropy which lie near the middle of the quasienergy spectrum.

In Sec. 5, we study a model in which the potential has a periodicity of 4 sites. The potential has an amplitude which is driven periodically in time and a phase $\phi$. The system has a mirror symmetry for two values of the phase, 0 and $7\pi/4$. At a DL point, a period-4 model with $\phi = 0$ behaves similarly as the period-2 model. But a period-4 model with $\phi = 7\pi/4$ exhibits a different and remarkably rich set of behaviors. First, in the absence of interactions, the Floquet Hamiltonian has the form of the Su-Schrieffer-Heeger (SSH) model in which the nearest-neighbor hoppings have a staggered structure; this leads to the appearance of modes near the ends of an open system. At a DL point, we obtain an extreme limit of the SSH model in which hoppings on alternate bonds vanish. This leads to a large number of conserved quantities which is the total fermion occupation number on two sites between which the hopping is non-zero. Labeling the unit cell with two such sites as $j$, the conserved occupation number $n_j$ can take the values 0, 1 and 2. It is convenient to map the two possible states of a unit cell with $n_j = 1$ to the states of a spin-1/2 object. We then discover that when interactions are introduced, a set of consecutive unit cells all of which have $n_j = 1$ is described by the transverse field Ising model, in which neighboring spin-1/2's have $\sigma_j^x \sigma_{j+1}^x$ interactions and there is a transverse magnetic field term $\sigma_j^z$. In addition, the two boundary sites of this model have a longitudinal magnetic field term $\sigma_j^x$. The exact spectrum of this model can be found by mapping the spin-1/2 model to a model of fermions using the Jordan-Wigner transformation. Once again we examine the spectrum of the entanglement entropy versus the quasienergy and the time evolution of the Loschmidt echo. We find a clear fragmentation of the Hilbert space in terms of the quasienergy spectrum, and the Loschmidt echo shows oscillations for a long period of time. Both of these are consequences of the conserved quantities. We then study the effects of a staggered on-site potential at a DL point; we find that the fragments of the Hilbert space further break up into secondary fragments. Finally, we study what happens in the period-4 model when both resonances and DL are simultaneously present. The Floquet Hamiltonian again consists of a density-dependent nearest-neighbor hopping. We summarize our main results in Sec. 7.

## 2 Hamiltonian of period-m model

In this paper we will discuss a class of periodically driven one-dimensional models of interacting spinless fermions. The general form of the Hamiltonian is

$$H(t) = \sum_{j=1}^{N} [J\,(c_j^\dagger c_{j+1} + \text{H.c.}) \,+\, \mu(t)\,\cos(2\pi j/m + \phi)\,c_j^\dagger c_j \,+\, V\,n_j n_{j+1}], \qquad (1)$$

where $J$ is a uniform time-independent nearest-neighbor hopping, $\mu(t)$ is the strength of an on-site potential which varies in space with period $m$ (we will call this a period-$m$ model), $\phi$ is a phase whose significance will be discussed later, $V$ is the strength of a nearest-neighbor density-density interaction, $n_j = c_j^\dagger c_j$, and $N$ denotes the number of sites (we will use periodic boundary conditions unless otherwise specified). We will take $\mu(t)$ to be a periodic function of time with a time period $T$ and the form

$$\begin{aligned} \mu(t) &= \mu\,f(t), \\ \text{where} \quad f(t) &= 1, \quad \text{for} \quad 0 \le t < T/2, \\ &= -1, \quad \text{for} \quad T/2 \le t < T, \end{aligned} \qquad (2)$$

and $f(t+T) = f(t)$. Since $H(t) = H(t+T)$, we will use the Floquet formalism to examine this model. The model can be analytically studied by performing a Floquet-Magnus expansion [1–13] which is valid in the large $\omega$ limit, where $\omega = 2\pi/T$ is the driving frequency. However, we will analytically study the model by finding an effective Floquet Hamiltonian $H_F$ using Floquet perturbation theory (FPT) [11, 73, 114–116]; this approach is valid when both $\omega$ and $\mu$ are much larger than all the other parameters of the model. We will examine in detail two classes of models, period-2 and period-4, both analytically using the Hamiltonian $H_F$ and numerically by computing the Floquet operator $U$ which evolves the system through one time period. We will set $\hbar = 1$ in this paper.

## 3 Period-2 model

In this section we will consider the period-2 model, whose form can be obtained by putting $m = 2$ and setting $\phi = 0$ in Eq. (1),

$$H(t) = \sum_{j=1}^{N} \left[ J\,(c_j^\dagger c_{j+1} + \text{H.c.}) \,+\, \mu(t)\,\cos(\pi j) \,+\, V\,n_j n_{j+1} \right]. \qquad (3)$$

This Hamiltonian can be written in the following form in the language of a unit cell with two sites,

$$H(t) = \sum_{j=1}^{N/2} \left[ J\,(a_j^\dagger b_j + a_j^\dagger b_{j-1} + \text{H.c.}) \,+\, \mu(t)\,(a_j^\dagger a_j - b_j^\dagger b_j) \,+\, V(n_{j,a} n_{j,b} + n_{j,a} n_{j-1,b}) \right], \quad (4)$$

where $N/2$ denotes the number of unit cells (we assume $N$ is even), and each unit cell consists of two sites labeled $a$ and $b$ with $a_j^\dagger$ and $b_j^\dagger$ denoting the creation operator for a particle at odd- and even-numbered sites, respectively. We will assume periodic boundary conditions.

We will first study the non-interacting case with $V = 0$. The Hamiltonian in momentum space is then given by

$$H(t) = \sum_{k} [J\,((1 + e^{-2ik})a_k^\dagger b_k + \text{H.c.}) \,+\, \mu(t)\,(a_k^\dagger a_k - b_k^\dagger b_k)], \qquad (5)$$

where $k$ takes $N/2$ equally spaced values lying in the range of $(-\pi/2, \pi/2]$. For each value of $k$, we define the Floquet operator

$$U_k = \mathcal{T} \exp[-i \int_0^T dt\, H_k(t)], \tag{6}$$

where $\mathcal{T}$ denotes time ordering. The Floquet operator can be written in the form

$$U_k = e^{-iTH_{Fk}}, \tag{7}$$

where $H_{Fk}$ is the time-independent effective Floquet Hamiltonian. Assuming $\mu \gg J$, we can write

$$\begin{aligned}
H_k(t) &= H_0(t) \,+\, H_1\,, \\
H_0 &= \mu(t)\,(a_k^\dagger a_k - b_k^\dagger b_k)\,, \\
H_1 &= \sum_k\, [J\,(1 + e^{-2ik})a_k^\dagger b_k + \text{H.c.}]\,.
\end{aligned} \tag{8}$$

We will now calculate $H_{Fk}^{(1)}$ using FPT to first order in the hopping $J$. We see from Eq. (2) that the two instantaneous eigenvalues of $H_0$ given by $E_k^\pm(t) = \pm\mu(t)$ satisfy the condition

$$e^{i\int_0^T dt\, [E_k^+(t) - E_k^-(t)]} \;=\; 1\,. \tag{9}$$

Hence we need to use degenerate FPT [11, 73, 114, 116].

The eigenfunctions corresponding to $E_k^\pm$ are given by

$$|+\rangle_k = \begin{pmatrix} 1 \\ 0 \end{pmatrix}, \quad \text{and} \quad |-\rangle_k = \begin{pmatrix} 0 \\ 1 \end{pmatrix}. \tag{10}$$

To construct the first-order Floquet Hamiltonian, we start with the Schrödinger equation

$$i\frac{d|\psi(t)\rangle}{dt} = (H_0 + H_1)|\psi(t)\rangle\,, \tag{11}$$

where we assume that $|\psi(t)\rangle$ has the form

$$|\psi(t)\rangle = \sum_n c_n(t) e^{-i\int_0^t dt'\,E_n(t')}|n\rangle\,, \tag{12}$$

and $|n\rangle = |\pm\rangle_k$ in our case. Using Eq. (11) and keeping terms up to first order in $H_1$, we find that

$$c_m(T) \;=\; c_m(0) - i\sum_n \int_0^T dt\, \langle m|H_1|n\rangle\, e^{i\int_0^t dt'\,[E_m(t') - E_n(t')]}\, c_n(0)\,. \tag{13}$$

This can be re-written as follows

$$c_m(T) = \sum_n (I - iH_{Fk}^{(1)}T)_{mn}\, c_n(0)\,, \tag{14}$$

where $I$ denotes the identity matrix and $H_{Fk}^{(1)}$ refers to the first-order effective Hamiltonian. We then find that

$$\begin{aligned}
\langle +|H_{Fk}^{(1)}|+\rangle &= 0\,, \quad \langle -|H_{Fk}^{(1)}|-\rangle \;=\; 0\,, \\
\langle +|H_{Fk}^{(1)}|-\rangle &= J\,(1 + e^{-2ik})e^{iA}\left(\frac{\sin A}{A}\right), \quad \langle -|H_{Fk}^{(1)}|+\rangle = J(1 + e^{2ik})e^{-iA}\left(\frac{\sin A}{A}\right),
\end{aligned} \tag{15}$$

where

$$A = \frac{\mu T}{2} = \frac{\pi \mu}{\omega}. \tag{16}$$

To first order in $H_1$, therefore, $H_F^{(1)}$ is given by

$$H_F^{(1)} = e^{iA} \left( \frac{\sin A}{A} \right) \sum_k \left[ J \left( 1 + e^{-2ik} \right) a_k^\dagger b_k + \text{H.c.} \right]. \tag{17}$$

Before discussing further, we note a symmetry of the Floquet operator,

$$[U_k(\mu, J)]^{-1} = U_k(\mu, -J), \tag{18}$$

which follows from the expression in Eq. (7) and the fact that the driving protocol satisfies $\mu(T - t) = -\mu(t)$. Eq. (18) implies [77] that

$$H_{Fk}(\mu, J) = - H_{Fk}(\mu, -J). \tag{19}$$

Hence $H_F$ can only have terms with odd powers of $J$. This implies that the next order term after the first order will be third order since there cannot be a term of second order in $J$. Hence, the first-order effective Hamiltonian will be a very good approximation to the exact Hamiltonian in the limit $\mu \gg J$.

We also note that the Floquet quasienergy $E_{Fk}$ must be an even function of $k$ if we hold $J$, $\mu$ fixed. To prove this, we do the unitary transformation

$$H_k(t) \rightarrow V_k H_k(t) V_k^{-1},$$
$$\text{where} \quad V_k = \begin{pmatrix} 1 & 0 \\ 0 & e^{-ik} \end{pmatrix}. \tag{20}$$

Thus we obtain

$$H_k(t) = \begin{pmatrix} \mu(t) & 2J \cos k \\ 2J \cos k & -\mu(t) \end{pmatrix}. \tag{21}$$

With this Hamiltonian, it is clear that

$$U_k(\mu, J) = U_{-k}(\mu, J), \quad \text{which implies that} \quad H_{Fk}(\mu, J) = H_{F,-k}(\mu, J). \tag{22}$$

Hence the Floquet quasienergy must be an even function of $k$.

## 3.1 Dynamical localization for a single-particle system

It is evident from the form of the Hamiltonian obtained by first-order FPT that the system will (approximately) exhibit DL [58–64] when

$$A = n\pi, \quad \text{i.e.,} \quad \mu = n\omega, \tag{23}$$

where $n$ is a non-zero integer. We note that this condition for DL becomes more and more exact as $\mu/J \rightarrow \infty$ and the higher order corrections to the first-order effective Hamiltonian become negligible. In this limit, $H_{Fk}^{(1)}$ vanishes for all values of $k$ which produces a flat band with zero quasienergy. We can see in Fig. 1 (a) that for a system with $J = 1$, $\mu = 20$, and $\omega = 20$, the quasienergy band is almost flat with a bandwidth $\Delta \sim 0.02$. In this case, we have taken $\mu \gg J$, and the higher order corrections to the first-order effective Hamiltonian are therefore very small. However, if we decrease the value of $\mu$ holding $J$ fixed, the first-order Floquet Hamiltonian becomes less and lass accurate, and the DL begins to fail as can be seen

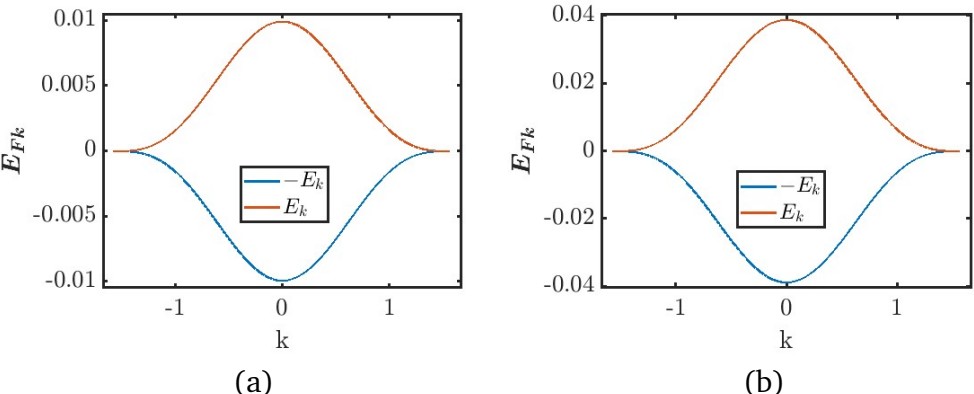

Figure 1: **Dispersion of quasienergies $E_{Fk}$ at DL points:** (a) Plot of $E_{Fk}$ versus $k$ obtained from the exact numerical calculation for $J = 1$, $\mu = 20$, and $\omega = 20$. We find that $\Delta \sim 0.02$, which implies an almost perfectly flat band at zero energy. (b) The same plot for $J = 1$, $\mu = 10$, and $\omega = 10$. In this case, the bandwidth $\Delta = 0.08$ which is four times larger than in the first case.

in Fig. 1 (b), for a system with $J = 1$, and $\mu = \omega = 10$. In this case, the bandwidth, $\Delta \sim 0.08$, which is four times larger than in Fig. 1 (a). This is due to the fact that the third-order effective Hamiltonian scales as $J^3/\mu^2$ at the dynamical localization points ($\mu = n\omega$) obtained from the first-order effective Hamiltonian as seen in Appendix C. Interestingly, the third-order effective Hamiltonian does not explicitly depend on the value of $\omega$ if one keeps $J$ and $\mu$ fixed at the DL points. Hence the bandwidth remains unaffected if $\omega$ is changed but $\mu = n\omega$ is kept fixed. For instance, we find numerically that the bandwidth is the same for the parameter values ($J = 1$, $\mu = 20$, $\omega = 20$) and ($J = 1$, $\mu = 20$, $\omega = 10$).

## 3.2  Dynamical phase transition

Motivated by our previous work [78], we now study the relaxation behavior of some correlators for the non-interacting model ($V = 0$). We will examine the correlation function $a_j^\dagger b_j$ (where $j$ denotes a particular unit cell) at stroboscopic times $t = nT$,

$$C_n = \langle \Psi_0 | a_j^\dagger(nT) b_j(nT) | \Psi_0 \rangle, \tag{24}$$

where $|\Psi_0\rangle$ is an initial state. For simplicity, we will take $|\Psi_0\rangle$ to be a product state in momentum space with the form

$$|\Psi_0\rangle = \prod_k a_k^\dagger |0\rangle. \tag{25}$$

Since this state is translation invariant, the correlator $C_n$ will not depend on the unit cell index $j$. We will now investigate the relaxation behavior of $C_n$ for large values of $n$, particularly to see if there is any crossover behavior. We observe numerically that generally in the $\mu \gg J$ limit (with a few exceptions), the correlation function exhibits a $n^{-1/2}$ decay with oscillations and there is no crossover behavior. To explain this, we first note that the Floquet quasienergy obtained from the first-order Floquet Hamiltonian has the form

$$E_{Fk}^\pm = \pm E_k,$$
$$\text{where} \quad E_k = 2J\left(\frac{\sin A}{A}\right)\cos k. \tag{26}$$

The stationary point of $E_k$ within the first Brillouin zone lies at $k = 0$. Next, we can show that the time-dependent part of the correlation function can be expressed as

$$\delta C_n \sim \frac{2}{N} \sum_k \left[ f(k) e^{-i2nTE_k} + \text{H.c.} \right], \tag{27}$$

where $f_k = \langle \Psi_0 | a_j^\dagger(0) b_j(0) | \Psi_0 \rangle$. For $N \to \infty$ and assuming $f(k)$ to be real, the above equation takes the integral form

$$\delta C_n \sim \frac{1}{\pi} \int_{-\pi/2}^{\pi/2} dk \, f(k) \, \cos(2nTE_k). \tag{28}$$

For large values of $n$, this integral gets a dominant contribution from the regions close to the stationary points of $E_k$ [75,77,78]. Therefore, expanding Eq. (28) around the stationary point at $k = 0$, we find that

$$\delta C_n \sim \frac{1}{\pi} \int dk \, f(k = 0) \, e^{i\zeta n(1 - k^2/2)}, \tag{29}$$

where $\zeta = 4JT(\sin A/A)$. Assuming $f(k) \neq 0$ for a generic initial state, we find that the correlation function for large values of $n$ will decay as a power $n^{-1/2}$, and there will be oscillations due to the term $\cos(\zeta n)$. This is the usual decay behavior unless there are competing terms which come from higher-order corrections.

We encounter such higher-order terms slightly away from the DL points (we recall that a DL point is where $A$ is an integer multiple of $\pi$, i.e., $\mu/\omega$ is an integer). Since an analytical calculation of the third-order effective Hamiltonian is a tedious task, we analyze this regime numerically. We first find the Floquet quasienergy from the numerically exact calculation and then do a fitting analysis of it. The results we obtain from such an analysis are as follows. We consider the parameter values $J = 1$, $\mu = 10$, and $\omega = 10.6$. Taking into account the structure of the stationary point obtained from the first-order effective Hamiltonian, we fit the numerically computed Floquet quasienergy around $k = 0$ as a function of $k$ up to sixth order in $k$. We find the following functional form

$$E(k) = p_4 k^4 + p_0, \quad \text{where} \quad p_4 = -0.02882, \quad p_0 = 0.07865. \tag{30}$$

All terms with odd powers of $k$ are found to be zero which is expected by the symmetry discussed in Eq. (22). Further, the coefficients of $k^2$ and $k^6$ are also found to be zero at these particular parameter values. Hence Eq. (30) shows that $E_k$ goes as $k^4$ near $k = 0$. An analysis similar to the one following Eq. (29) then shows that

$$\delta C_n \sim \frac{1}{\pi} \int dk \, f(k = 0) \, e^{i2nT(p_0 + p_4 k^4)}. \tag{31}$$

Assuming $f(k) \neq 0$, we see that for large values of $n$, the correlation function will decay as $n^{-1/4}$ with oscillations due to the $\cos(2nTp_0)$ term. This is what we see in the Fig. 2 (c): for a system with the parameter values mentioned above, the correlation function function decays as an oscillatory term times $n^{-1/4}$. If we plot $|\delta C_n|$ (rather than $\delta C_n$) versus $n$, the period of oscillations $\Delta n$ will be given by the condition, $2\Delta n p_0 T = \pi$, which implies that $\Delta n = \omega/(4p_0)$. Putting $\omega = 10.6$, and $p_0 = 0.07865$, we find that $\Delta n \sim 34$), which agrees very well with the oscillation period seen in Fig. 2. Interestingly, we observe a crossover from $n^{-1/4}$ to $n^{-1/2}$ as we move slightly away from $\omega = 10.6$. As mentioned earlier, the correlation function in general decays as a power $n^{-1/2}$ for large values of $n$ in this class of systems. However, we observe a different power law decay behavior ($n^{-1/4}$) emerging at $\omega \simeq 10.6$, and we, therefore, call

$\omega \simeq 10.6$ the critical frequency $\omega_c$. To see the crossover, we consider $J = 1$, $\mu = 10$, and $\omega = 10.7$ as an example. We again follow the same fitting procedure, and find the following functional form

$$E(k) = p_4 k^4 + p_2 k^2 + p_0, \text{ where } p_4 = -0.02823, \; p_2 = -0.009116, \; p_0 = 0.0981. \quad (32)$$

The terms with odd powers of $k$ vanish as before. In contrast to the previous case, however, there is now a competition between the $k^2$ and $k^4$ terms, which can be seen in Eq. (32). Close to $k = 0$, we see that $E_k$ has a leading contribution coming from the $k^4$ term, followed by a subleading correction due to the $k^2$ term very close to $k = 0$. Expanding the integrand in Eq. (28) around $k = 0$ then gives

$$\delta C_n \sim \frac{1}{\pi} \int dk \, f(k=0) \, e^{i 2nT(p_0 + p_2 k^2 + p_4 k^4)}. \quad (33)$$

Defining a scaled variable $k' = k n^{1/4}$, and assuming $f(k) \neq 0$, we find

$$\delta C_n \sim \frac{1}{\pi n^{1/4}} e^{i 2nT p_0} f(k=0) \int dk' \, e^{i T(p_2 k'^2 n^{1/2} + p_4 k'^4)}. \quad (34)$$

Since $|p_2| \ll |p_4|$, it is clear from Eq. (33) that there will be a $n^{-1/4}$ scaling (along with oscillations) when $|\epsilon'| n^{1/2} \ll 1$, where $\epsilon' = p_2/p_4$. However, when $|\epsilon'| n^{1/2} \gg 1$, the $n^{-1/4}$ scaling breaks down and we then encounter a different scaling law, namely, $n^{-1/2}$ (along with oscillations due to the $\cos(2nT p_0)$ term). We can extract the crossover scale $n_c$ from this analysis; a crossover between the $n^{-1/4}$ and $n^{-1/2}$ power-laws occurs when $|\epsilon'| n_c^{1/2} \sim 1$, which implies that $n_c \sim 1/|\epsilon'|^2$. To see this behavior, we define $\epsilon$ as $\omega = 1/(1/\omega_c - \epsilon)$ [78], where $\epsilon \propto \epsilon'$, and we look at the divergence behavior near $\omega_c$. We plot the crossover scale $n_c$ with $\epsilon$ in Fig. 2 (e), and then numerically fit the plot of $n_c$ versus $\epsilon$. We find that $n_c \sim 1/|\epsilon|^2$ which agrees with the analytically derived result.

## 3.3 Effects of interactions on dynamical localization

In this section, we will look at the effects of density-density interactions [61, 63], $H_I = \sum_j V n_j n_{j+1}$ on DL. For this case, we take the system to be at half-filling with periodic or antiperiodic boundary conditions ($c_{N+1} = c_1$ or $-c_1$) depending on whether the particle number is even or odd, respectively, to avoid any degeneracy in the spectrum. To get an analytical understanding of this system, we will first compute the effective Hamiltonian up to first order in $V$. Since $H_I$ is diagonal in the position basis and commutes with the unperturbed Hamiltonian, $H_0(t)$, the effective Hamiltonian to first order in $V$ simply reads as

$$H_{FI}^{(1)} = \sum_j V n_j n_{j+1}. \quad (35)$$

Hence the full effective Hamiltonian to first order in $J$ and $V$ is as follows

$$\begin{aligned} H_F^{(1)} &= H_{F,J}^{(1)} + H_{FI}^{(1)}, \\ H_{F,J}^{(1)} &= e^{iA} \left( \frac{\sin A}{A} \right) \sum_k \left[ J(1 + e^{-2ik}) a_k^\dagger b_k + \text{H.c.} \right], \\ H_{FI}^{(1)} &= \sum_j V n_j n_{j+1}. \end{aligned} \quad (36)$$

We note that the Floquet evolution operator $U(T)$ satisfies the same condition as mentioned in Eq. (19) [77], and therefore $H_F$ possesses the symmetry

$$H_F(\mu, J, V) = -H_F(\mu, -J, -V). \quad (37)$$

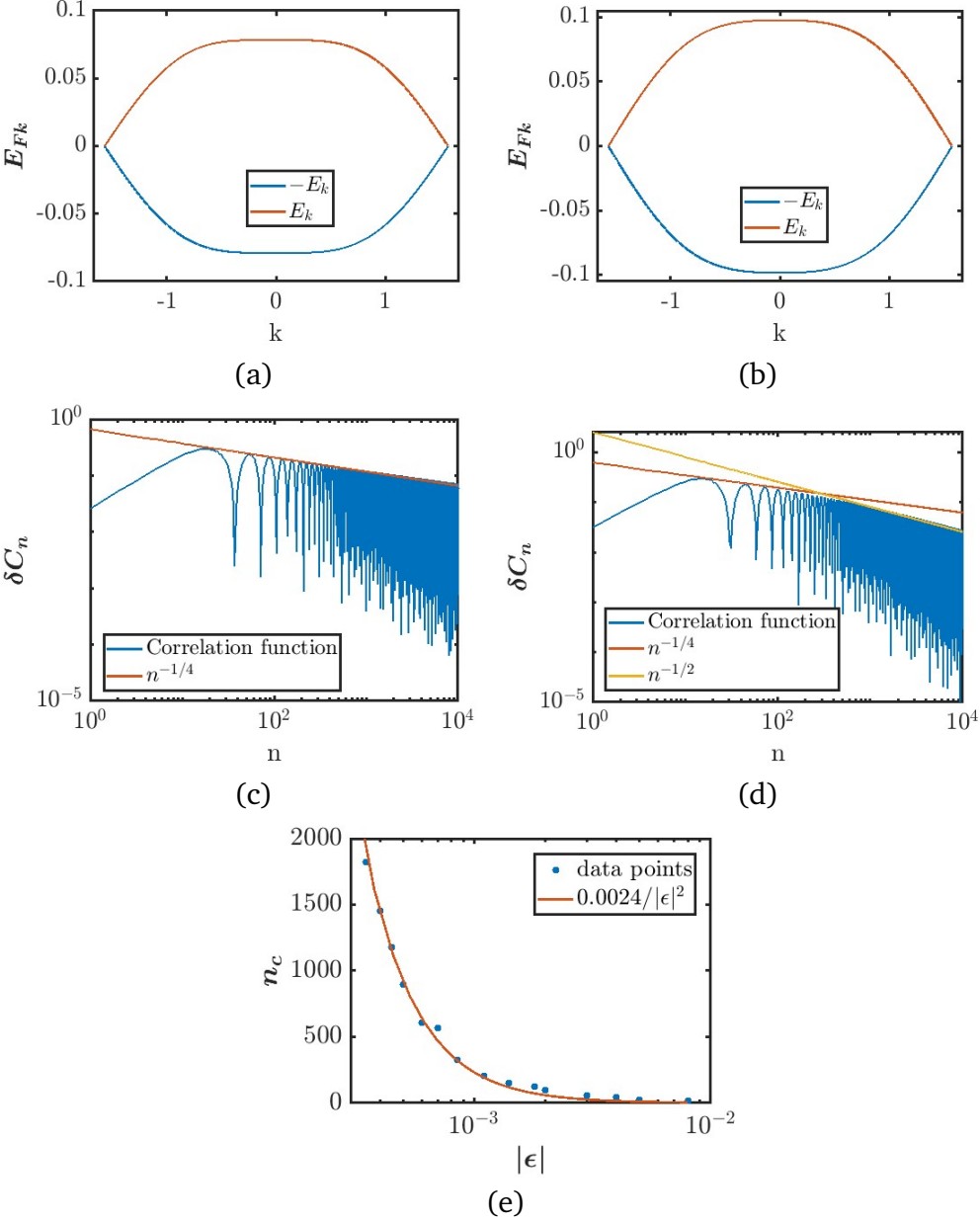

Figure 2: **Quasienergies and crossover behaviors of correlation functions:** Plots showing $E_{Fk}$ as a function of $k$ obtained from the exact numerical calculation for $J = 1$, $\mu = 10$, and (a) $\omega = 10.6$ and (b) $\omega = 10.7$. In plot (a), $E_k \sim k^4$ around $k = 0$ as can be seen in Eq. (31). In plot (b), $E_k \sim k^2 + \epsilon k^4$, where $|\epsilon| \ll 1$, as seen in Eq. (33). Log-log plots of the absolute value of the $n$-dependent part of the correlation function $\delta C_n$ as a function of the time $nT$, for $J = 1$, $\mu = 10$, and (c) $\omega = 10.6$ and (d) $\omega = 10.7$. (c) The correlation function decays as $n^{-1/4}$ along with oscillations. (d) The plot shows a crossover between an oscillatory term times $n^{-1/4}$ and an oscillatory term times $n^{-1/2}$. (e) Plot showing the variation of $n_c$ with $|\epsilon|$ as we approach the critical frequency from the $\omega > \omega_c$ side, where $\omega_c \simeq 10.6$ is the frequency where the correlation function decays as $n^{-1/4}$. The numerically obtained fitting indicates that $n_c \sim 1/|\epsilon|^2$.

Hence the first-order effective Hamiltonian will be a very good approximation to the exact Hamiltonian for $\mu \gg J, V$, since the higher order corrections will be negligible compared to the first-order term. We see from Eq. (36) that $H_{F,J}^{(1)} = 0$ at the DL points where $A$ is an integer multiple of $\pi$, and then $H_F^{(1)}$ just reduces to $H_{FI}^{(1)}$. Consequently, the spectrum of the Floquet quasienergies becomes easy to compute at any filling due to the diagonal form of the effective Hamiltonian in the position basis.

We now consider a system at half-filling with antiperiodic boundary conditions for $J = 1$, $\mu = 20$, $\omega = 20$, $V = 1$, and $L = 16$, and calculate the spectrum of the Floquet quasienergies and half-chain entanglement entropy. The dimension of the Hilbert space for $N = L/2$ and $L = 16$ is $^{16}C_8 = 12870$. We first discuss the Floquet quasienergies and the degeneracies for some of the Floquet eigenstates which can be obtained analytically. We observe that there are eight equally spaced quasienergies lying between 0 and $8V$ with a energy spacing of $V$ for $L = 16$, as shown in Fig. 3 (a). Next, there are exactly two Floquet eigenstates, $|\pm\rangle = 1/\sqrt{2}\left(|1'\rangle \pm |2'\rangle\right)$ with $E_F = 0$, where $|1'\rangle$ and $|2'\rangle$ are equal to $|1010101010101010\rangle$ and $|0101010101010101\rangle$ respectively in the number basis. Furthermore, there are 16 Floquet eigenstates with $E_F = 8V$. We note that the DL induces an emergent integrability which leads to the appearance of many flat bands and several low-entanglement states (with $S_{L/2} \ll S_{page}$, where $S_{page} = (L/2) \ln 2 - 1/2$ [117, 118]) near the middle of the spectrum as can be seen in Figs. 3 (a) and (b) respectively. However, this emergent integrability starts to break down as we move away from a DL point, as we see in Fig. 3 (d), for $J = 1$, $\mu = 20$, $\omega = 22$ and $V = 1$. In Figs. 3 (c) and (d), we observe that the flatness of the bands begins to disappear as we move away from the DL limit.

In Fig. 4, we show some dynamical properties of the system, namely, the two-point correlation function and the Loschmidt echo at a DL point and away from a DL point. The parameter values chosen for this figure are the same as in Fig. 3. We will take the initial state to be the ground state of the undriven Hamiltonian. In Fig. 4 (a), we see almost perfect revivals of the correlation function in time, which is expected due to the emergent integrability at a DL point. Further, the oscillation period of the revivals can be calculated by using the condition $e^{iV\delta t} = 1$. Hence the time period is given by $\delta t = 2\pi/V$, where $V = 1$ in our case. Away from the DL point, the correlation function decays rapidly in time due to a breakdown of the integrable structure. In Fig. 4 (c), we show the Loschmidt echo, $|\langle\psi(t)|\psi(0)\rangle|$, for the same parameter values as in Figs. 4 (a) and 4 (b) and with the same initial states. For the first case with $\omega = 20$, we again see perfect revivals with a period of $2\pi$ in Fig. 4 (c), upper panel. For the second case with $\omega = 22$, the amplitude of Loschmidt echo decays rapidly as shown in Fig. 4 (c), lower panel. These results indicate that the system evades thermalization for a long time at a DL point but thermalizes quickly as we go away from a DL point [58–64].

## 3.4 Effects of resonances

In this section we will examine the effects of resonances [5, 113] on DL for the period-2 model. To obtain an analytical insight for this case, we will consider parameter values with $\mu = \omega = V \gg J$ and derive an effective Floquet Hamiltonian using first-order FPT. We first consider a four-site system so that we can easily identify various non-trivial processes, and we will then generalize it to larger system sizes. We find that a four-site system only offers four distinct non-trivial processes for a particular choice of a periodic potential pattern due to the constraints imposed by DL. We further note that there are two possible potential patterns available for such a system consisting of four sites. Therefore, we need to consider a total of eight distinct non-trivial processes while formulating the first-order effective Floquet Hamiltonian. These eight processes and the corresponding time-dependent effective Hamiltonians are listed below.

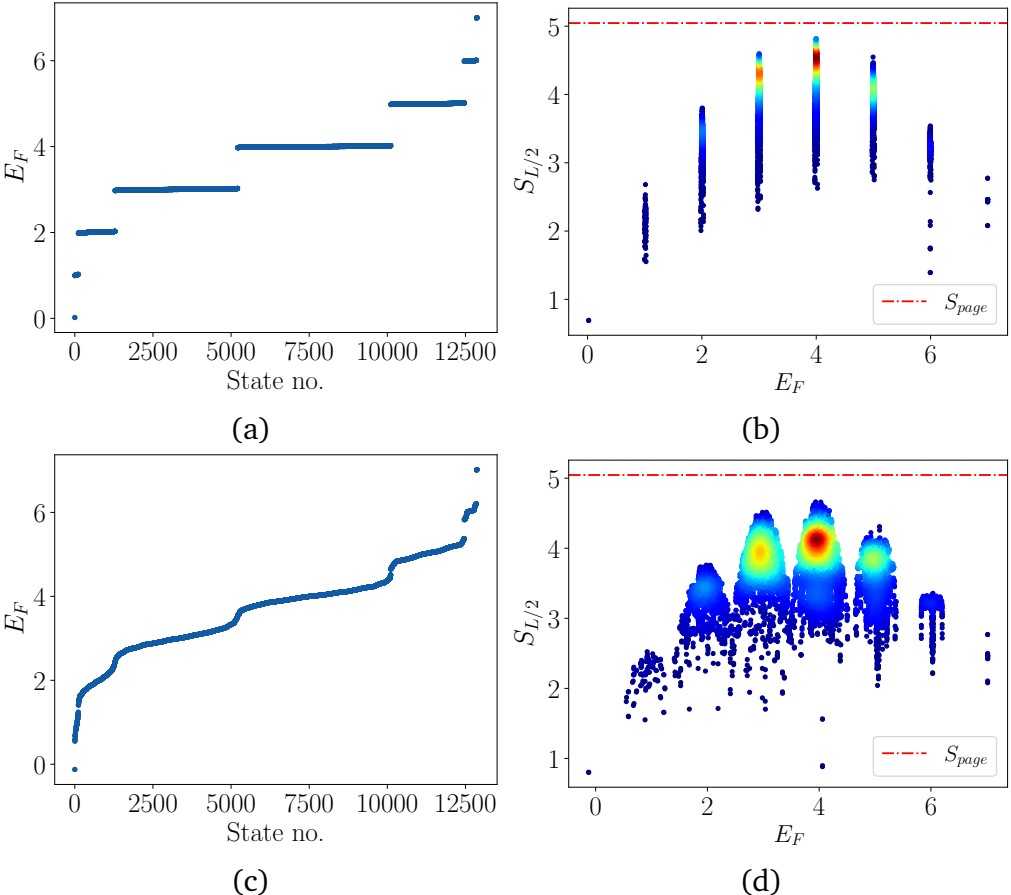

Figure 3: **Quasienergy and entanglement entropy spectrum of the period-2 model at a DL point and away from a DL point:** (a-b) Plots of the quasienergy spectrum $E_F$ and half-chain entanglement entropy $S_{L/2}$ as a function of $E_F$ exactly at a dynamical localization point with $J = 1$, $\mu = \omega = 20$, and $V = 1$, where the system exhibits many-body flat bands with many low-entanglement states near the middle of the spectrum. (c-d) Plots showing the same quantities as in plots (a-b) but away from a DL point, with $J = 1$, $\mu = 20$, $\omega = 22$ and $V = 1$. For this case, the many-body flat bands start to disappear as we tune the system away from a DL point.

As an example, we consider the first process listed above and calculate the first-order Floquet Hamiltonian for this case. We first note that the Hamiltonian can be recast as

$$
\begin{aligned}
H(t) &= (\mu(t) + V/2)\, I \;-\; (\mu - V/2)\, \sigma^z \;+\; J\, \sigma^x\,, \\
H_0 &= -(\mu - V/2)\, \sigma^z\,, \\
H_1 &= J\, \sigma^x\,,
\end{aligned}
\tag{38}
$$

where $I$, $\sigma^x$ and $\sigma^z$ denote the identity and two of the Pauli matrices, and $H_0$ and $H_1$ are the unperturbed Hamiltonian and perturbation, respectively. Assuming $\mu = \omega$ and $V \gg J$, the instantaneous eigenvalues of $H_0$ are given by $E_k^{\pm} = \pm(\mu(t) + V/2)$. The eigenfunctions corresponding to $E_k^{\pm}$ are given by $|+\rangle = \begin{pmatrix} 0 \\ 1 \end{pmatrix}$ and $|-\rangle = \begin{pmatrix} 1 \\ 0 \end{pmatrix}$. These two eigenvalues

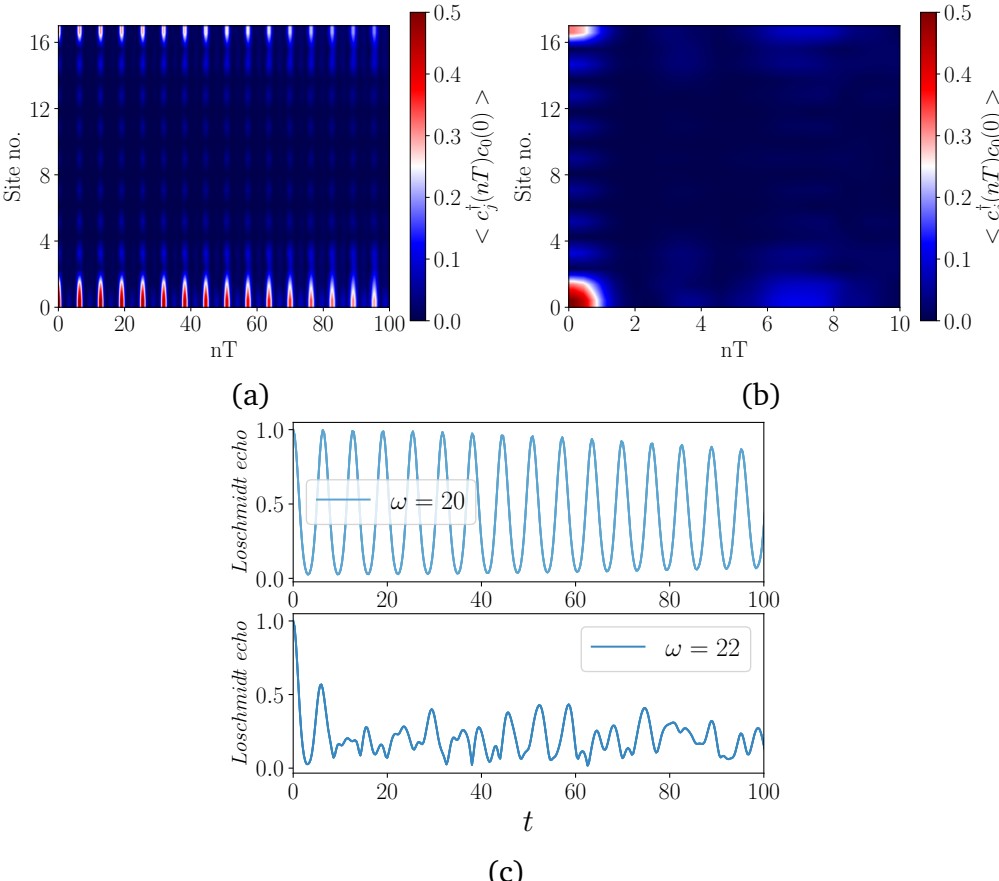

Figure 4: **Dynamics of the correlation function and Loschmidt echo at a DL point and away from a DL point for the period-2 model:** (a-b) Surface plots showing the two-point correlation function as a function of site number and time $nT$ with $n$ being the driving cycle number, at a dynamical localization point and away from a dynamical localization point. The parameter values chosen for plots (a) and (b) are the same as in Figs. 3. (c): Plots showing the Loschmidt echo versus time for the same parameter values as in Figs. 3. For all four cases, we choose the initial state to be the ground state of the undriven Hamiltonian. As shown in (a) and the upper panel of (c), the dynamics demonstrates long-time oscillatory behaviors indicating a non-ergodic behavior at a DL point. However, both correlation function and Loschmidt echo decay rapidly with time as we move away from a DL point, as can be seen in (b) and the lower panel of (c).

satisfy the condition given in Eq. (9), and we therefore use degenerate FPT. This gives

$$\langle +|H_F^{(1)}|+\rangle = 0\,, \quad \langle -|H_F^{(1)}|-\rangle = 0\,,$$

$$\langle +|H_F^{(1)}|-\rangle = J\, I(\mu,V,T)\,, \quad \langle -|H_F^{(1)}|+\rangle = J\, I^*(\mu,V,T)\,,$$

$$I(\mu,V,\omega) = \frac{e^{-i(2\mu-V)T/4}\sin((2\mu-V)T/4)}{(2\mu-V)T/2} + \frac{e^{-i(2\mu-3V)T/4}\sin((2\mu+V)T/4)}{(2\mu+V)T/2}\,. \tag{39}$$

Putting $\mu = V = \omega$, we find that $I(\mu,V,\omega) = -4i/(3\pi)$. Thus, the effective Hamiltonian for this particular process is

$$H_F^{(1)} = -\frac{4i}{3\pi}\, n_0 c_2^\dagger c_1 (1-n_3)\, +\, \text{H.c.}\,, \tag{40}$$

Table 2: Allowed processes and their corresponding effective time-dependent Hamiltonians for a four-site system with all possible patterns of periodic on- site potential in the case dynamical localization and resonance for a period-2 model.

| Pattern of periodic potential | Process | Effective time-dependent Hamiltonian |
|---|---|---|
| + - + - | $1100 \leftrightarrow 1010$ | $H(t) = \begin{pmatrix} V & J \\ J & 2\mu(t) \end{pmatrix}$ |
| + - + - | $0100 \leftrightarrow 0010$ | $H(t) = \begin{pmatrix} -\mu(t) & J \\ J & \mu(t) \end{pmatrix}$ |
| + - + - | $0101 \leftrightarrow 0011$ | $H(t) = \begin{pmatrix} -2\mu(t) & J \\ J & V \end{pmatrix}$ |
| + - + - | $1101 \leftrightarrow 1011$ | $H(t) = \begin{pmatrix} V-\mu(t) & J \\ J & V+\mu(t) \end{pmatrix}$ |
| - + - + | $1100 \leftrightarrow 1010$ | $H(t) = \begin{pmatrix} V & J \\ J & -2\mu(t) \end{pmatrix}$ |
| - + - + | $0100 \leftrightarrow 0010$ | $H(t) = \begin{pmatrix} \mu(t) & J \\ J & -\mu(t) \end{pmatrix}$ |
| - + - + | $0101 \leftrightarrow 0011$ | $H(t) = \begin{pmatrix} 2\mu(t) & J \\ J & V \end{pmatrix}$ |
| - + - + | $1101 \leftrightarrow 1011$ | $H(t) = \begin{pmatrix} \mu(t)+V & J \\ J & V-\mu(t) \end{pmatrix}$ |

where we have set $J = 1$. Following similar procedures, we can compute the effective Hamiltonians for all the other processes. These are given below.

Taking all these processes into account, the complete effective Hamiltonian for the case where a resonance and DL occur simultaneously is given by

$$
H = -\frac{4i}{3\pi} \sum_{j=1}^{L/2} \left[ (1-n_{2j})c^{\dagger}_{2j+2}c_{2j+1}n_{2j+3} + n_{2j}c^{\dagger}_{2j+2}c_{2j+1}(1-n_{2j+3}) + \text{H.c.}\right]
$$
$$
+ \frac{4i}{3\pi} \sum_{j=1}^{L/2}\left[ n_{2j+1}c^{\dagger}_{2j+3}c_{2j+2}(1-n_{2j+4}) + (1-n_{2j+1})c^{\dagger}_{2j+3}c_{2j+2}n_{2j+4} + \text{H.c.}\right]. \tag{41}
$$

We can perform the unitary transformation $c_{2j} \to c_{2j}$ and $c_{2j+1} \to ic_{2j+1}$ to obtain a simpler form of the effective Hamiltonian

$$
H = \frac{4}{3\pi} \sum_{j=1}^{L} (n_j - n_{j+3})^2 \left( c^{\dagger}_{j+2}c_{j+1} + \text{H.c.}\right). \tag{42}
$$

This form implies that hoppings between two nearest-neighbor sites are forbidden whenever their neighboring sites are both completely empty or completely occupied. Hence, these forbidden processes act as kinetic constraints in the dynamics [105–110], and these constraints can, in principle, lead the system towards an anomalous thermalization behavior. We also find that the Hamiltonian in Eq. (42) has several zero-energy states which consist of single states in the number basis. The number of such states can be found using a transfer matrix method as shown in Appendix A. We discover that the number grows exponentially with system size as $1.466^L$.

This mechanism can be further generalized to a lower frequency regime by considering the other DL points and resonances given by $\mu = V = n\omega$ ($n \neq 1$), keeping $\mu, V >> J$. To obtain an analytical insight in this limit, we can derive the first-order FPT Hamiltonian following

Table 3: First-order effective FPT Hamiltonians for the allowed processes with all possible patterns of periodic potential in the case of dynamical localization and resonance for a period-2 model.

| Pattern of periodic potential | Process | First-order Floquet Hamiltonian |
|---|---|---|
| + - + - | $1100 \leftrightarrow 1010$ | $H_F^{(1)} = -\frac{4i}{3\pi} n_0 c_2^\dagger c_1 (1-n_3) +$ H.c. |
| + - + - | $0100 \leftrightarrow 0010$ | $H_F^{(1)} = 0$ |
| + - + - | $0101 \leftrightarrow 0011$ | $H_F^{(1)} = -\frac{4i}{3\pi} (1-n_0) c_2^\dagger c_1 n_3 +$ H.c. |
| + - + - | $1101 \leftrightarrow 1011$ | $H_F^{(1)} = 0$ |
| - + - + | $1100 \leftrightarrow 1010$ | $H_F^{(1)} = \frac{4i}{3\pi} n_0 c_2^\dagger c_1 (1-n_3) +$ H.c. |
| - + - + | $0100 \leftrightarrow 0010$ | $H_F^{(1)} = 0$ |
| - + - + | $0101 \leftrightarrow 0011$ | $H_F^{(1)} = \frac{4i}{3\pi} (1-n_0) c_2^\dagger c_1 n_3 +$ H.c. |
| - + - + | $1101 \leftrightarrow 1011$ | $H_F^{(1)} = 0$ |

a similar procedure as charted out before. Interestingly, the first-order FPT Hamiltonian for $\mu = V = n\omega$, where $\mu$, $V \gg J$, and $n$ is odd, reads as

$$H_F^{(1)} = \frac{4}{3\pi n} \sum_{j=1}^{L} (n_j - n_{j+3})^2 \left( c_{j+2}^\dagger c_{j+1} + \text{H.c.} \right), \tag{43}$$

which is the same Hamiltonian as obtained for the case $\mu = V = \omega$, but with a hopping strength $4/(3\pi n)$. On the other hand, the first-order effective Hamiltonian for $\mu = V = n\omega$, where $\mu$, $V \gg J$ and $n$ is *even*, turns out to be

$$H_F^{(1)} = 0, \tag{44}$$

which implies that there should be a many-body flat band lying at $E_F = 0$.

In Figs. 5 (a) and (b), we show the variation of the half-chain entanglement entropy $S_{L/2}$ as a function of $E_F$, obtained from exact numerical calculations and from the first-order FPT Hamiltonian shown in Eq. (42) for $J = 1$, $\mu = \omega = 20$, and $V = 20$. Both these cases point towards many low-entanglement states near the middle of the spectrum; these arise due to the kinetic constraints simultaneously imposed by DL and the resonance condition. Before proceeding further, we note that the effective Hamiltonian described in Eq. (42) supports many fragmented Hilbert space sectors [105–110], which can be shown as follows. First, there are an exponentially large number of fragments each of which consists of a single state with zero energy; this is shown in Appendix A. Next, there are simple fragments consisting of only four states. Consider, for example, the fragment containing the states $|0000111111110000\rangle$, $|0001011111110000\rangle$, $|0000111111101000\rangle$, and $|0001011111101000\rangle$, and their translated partners (we have written all the states in the occupation number basis). The action of the effective Hamiltonian on these four states is schematically shown in Fig. 6. Taking into account the action of the effective Hamiltonian on these four states, we find an effective $4 \times 4$ Hamiltonian which represents this particular fragment,

$$H_{frag} = \frac{4}{3\pi} \begin{pmatrix} 0 & 1 & 1 & 0 \\ 1 & 0 & 0 & 1 \\ 1 & 0 & 0 & 1 \\ 0 & 1 & 1 & 0 \end{pmatrix}. \tag{45}$$

The eigenvalues of this Hamiltonian are given by $E_{1,2} = \pm 0.85$, $E_3 = 0$, and $E_4 = 0$. Hence these four eigenvalues offer two distinct difference in energies, i.e., $\Delta E = 0.85$ and $\Delta E = 1.7$, which will be important later in the discussion of dynamics.

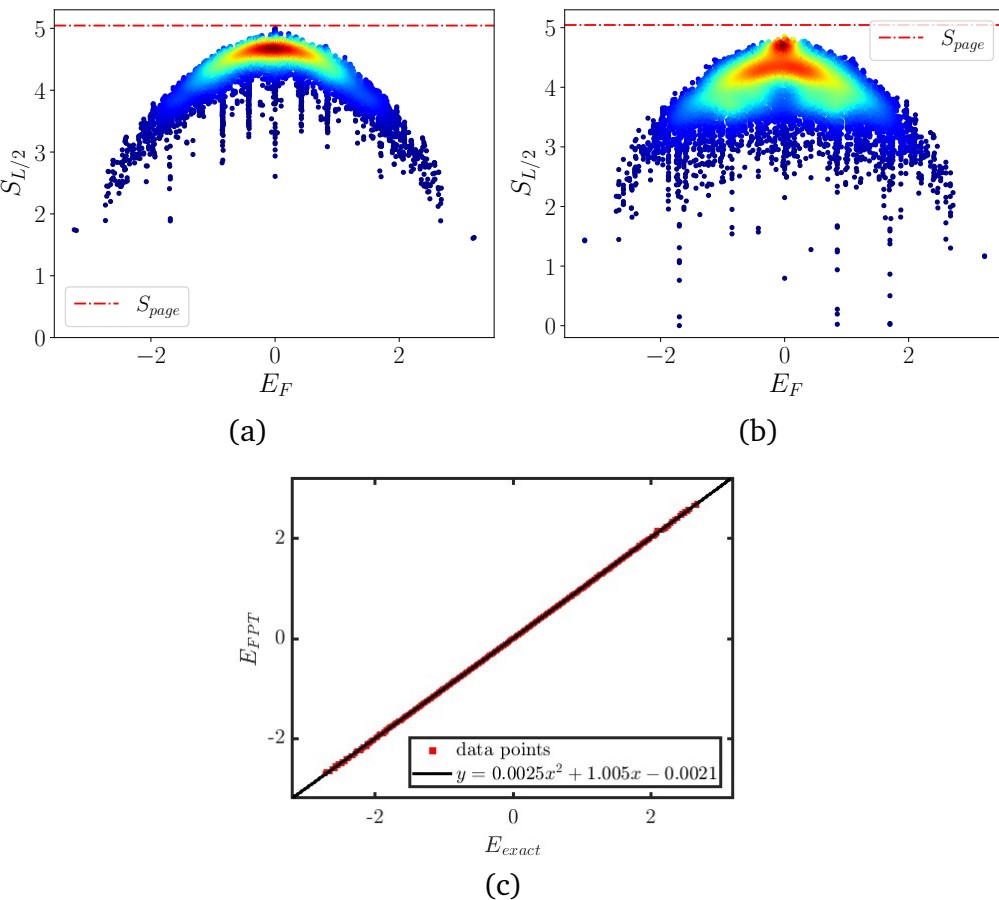

Figure 5: **Entanglement entropy spectrum of the period-2 model for the case of DL and resonance:** Plots showing the entanglement entropy $S_{L/2}$ as a function of the Floquet quasienergy $E_F$ obtained from (a) the exact numerical calculation and (b) the first-order effective FPT Hamiltonian shown in Eq. (42), for $J = 1$, $\mu = \omega = V = 20$, and $L = 16$. (c) Plot showing $E_{exact}$ versus $E_{FPT}$ obtained numerically for the same parameter values as in plots (a) and (b). In (a) and (b), we see many low-entanglement states near the middle of the spectrum. In both plots, the color intensity indicates the density the states, implying that the majority of Floquet eigenstates show almost thermal entanglement. Plot (c) shows that the quasienergies obtained from the first-order FPT agree quite well with the exact numerically computed values. However, as plots (a) and (b) show, there are a large number of Floquet eigenstates for which the entanglement estimated from FPT is much smaller than the exact numerically obtained values.

In Figs. 7 (a) and (b), the variation of the Loschmidt echos with time, $t = nT$, is shown as found from the exact numerical calculation and the first-order FPT, respectively, for the parameter values, $J = 1$, $\mu = \omega = V = 20$, and $L = 16$, taking the initial state to be $|0000111111110000\rangle$. We can show analytically that the Loschmidt echo for this particular choice of initial state takes the form $|a + b\cos(\Delta E t)|$, which implies that it oscillates with a period $\Delta t = 2\pi/\Delta E$. Putting $\Delta E = 0.85$, the period of oscillation in the revival pattern turns out to be $\Delta t \simeq 7.4$, which almost perfectly captures the numerically obtained value. In Fig. 7 (c), we show the overlaps of the same initial state with the Floquet eigenstates (obtained from the exact numerical calculation) as a function of $E_F$, where the color bar indicates the variation of $S_{L/2}$ of the Floquet eigenstates. Interestingly, we observe that the overlap is highest



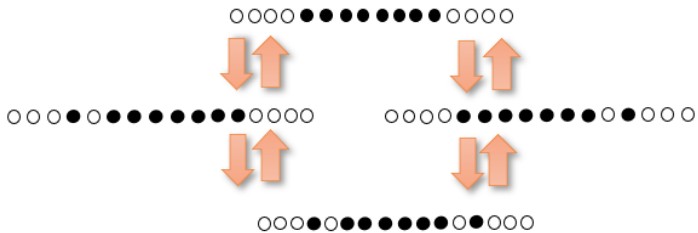

Figure 6: **Schematic of a Hilbert space fragment for the period-2 case:** Figure showing a particular Hilbert space sector consisting of four states, $|0000111111110000\rangle$, $|0001011111110000\rangle$, $|0000111111101000\rangle$, and $|0001011111101000\rangle$. The black and white dots indicate occupied and empty sites respectively.

for the Floquet eigenstates with $E \simeq \pm 0.85$ and 0, which almost identically agrees with the analytically predicted values. Similarly in Figs. 8 (a-b), we show the entanglement entropy as a function of the quasienergy for $\omega \approx 6.67$ (third DL point, i.e., $n = \mu/\omega = 3$) and $\omega \approx 2.22$, (ninth DL point, i.e., $n = 9$) with the rest of the parameters being the same as in Fig. 5. In both cases, we see many low-entanglement states with the range of quasienergies approximately being 1/3 and 1/9 times those of the first case shown in Fig. 5; this agrees with the analytically derived first-order FPT Hamiltonian shown in Eq. (43). In Fig. 9, we again examine the dynamics of the Loschmidt echo for $J = 1$, $\mu = V = 20$, $L = 16$, and $\omega \approx 6.67$, 4, 2.85, and 2.22, which correspond to the third, fifth, seventh, and ninth DL points. [Note that the last value of $\omega$ is not much larger than the hopping amplitude $J$, and therefore does not lie in the high-frequency regime. Nevertheless we see that the Loschmidt echo decays very slowly. This shows that DL and resonances lead to very slow thermalization even when the driving frequency is not very large]. For all four cases, we consider the same initial state as before, and we see that the Loschmidt echo demonstrates long-time revivals, indicating that the system shows very slow thermalization. Furthermore, the period of oscillations in the revival pattern for all four cases can be explained by considering the first-order FPT Hamiltonian and the effective Hamiltonian for the HSF cluster consisting of the four states described in Fig. 6. As shown in Eq. (43), the DL points for $\mu = V = n\omega$ ($n = 3$, 5, 7, $\cdots$) renormalizes the effective hopping strength of the effective Hamiltonian obtained for the case with $\mu = V = \omega$ by a factor of $1/n$. We can then argue that the period of oscillations in the revival pattern for these four cases will be $\Delta t_n = n\, 2\pi/\Delta E$, where $n = 3$, 5, 7,... and $\Delta E \approx 0.85$. This implies that the period of oscillations corresponding to the third, fifth, seventh and ninth DL points will be $\Delta t_n \approx 22.2$, 37, 51.8, and 66.6, which almost perfectly agrees with the exact numerical calculation as seen in Fig. 9.

## 4 Thermodynamic stability of Hilbert space fragmentation in period-2 model

In this section, we will discuss the stability of HSF in the thermodynamic limit ($L \to \infty$) in the context of the period-2 model which is at a DL point along with a resonance. In Figs. 10 (a-c), the scaled entanglement entropy $S_{L/2}/L$ is shown as a function of the scaled quasienergy $E_F/L$ for $L = 12$, 16 and 20, respectively, for $\mu = V = \omega = 20$. For this analysis, we employed the exact diagonalization method for individual momentum sectors by using the translation

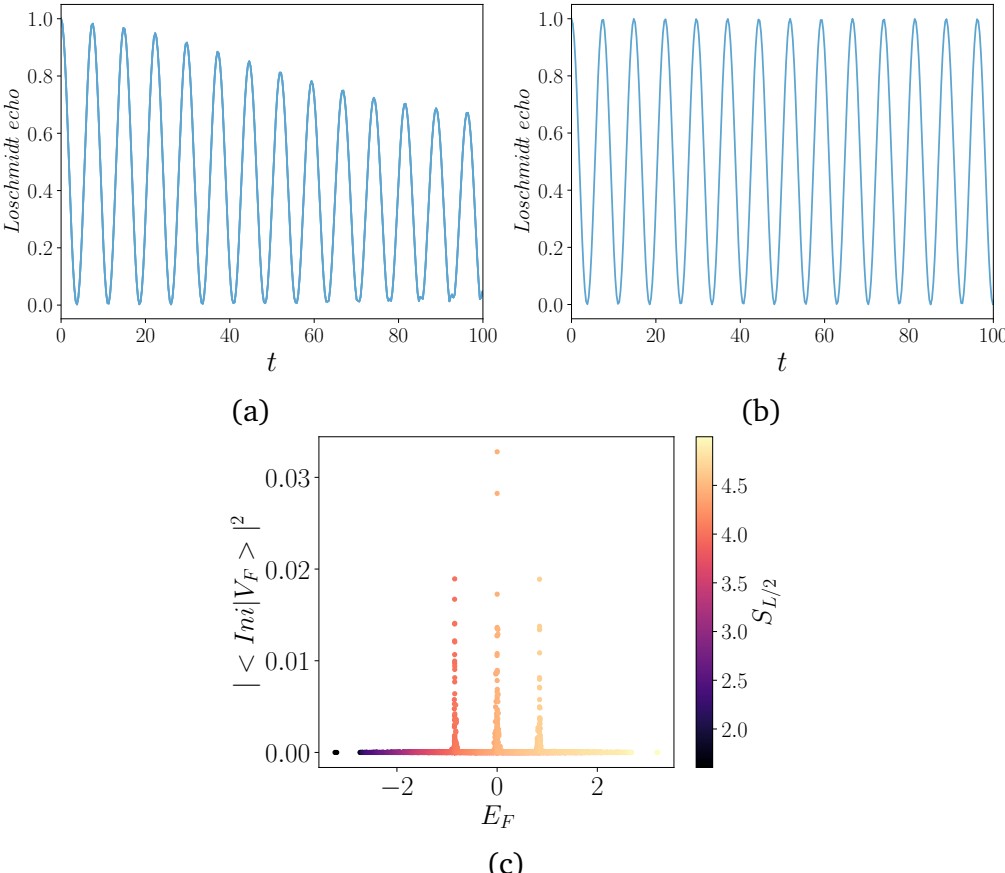

Figure 7: **Dynamics of the Loschmidt echo and overlaps with Floquet eigenstates in a resonant case:** (a-b): Plots of the Loschmidt echo versus time as obtained from (a) the exact numerical calculation and (b) the first-order effective FPT Hamiltonian, for the same parameter values as in Fig. 5, taking the initial state ($|Ini\rangle$) to be $|0000111111110000\rangle$. Both plots show show long-time oscillations in time showing an anomalous thermalization behavior. (c): Overlaps of the same initial state with the Floquet eigenstates as a function of $E_F$ computed from the exact numerical calculation for the same parameter values, with a color bar indicating the variation of $S_{L/2}$. The quasienergies of the Floquet eigenstates having the highest overlaps with the initial state agree with the analytically predicted values.

symmetry in order to access larger system sizes. For all three values of $L$, the range of the scaled quasienergy turns out to be the same since the many-body bandwidth increases linearly with $L$. Interestingly, we find that the scaled entanglement spectrum broadens with increasing system size, as can be seen from Figs. 10 (a-c). The broadening of the entanglement spectrum indicates an increasing number of Hilbert space fragments and inert configurations of states for larger system sizes. In Figs. 11 (a-b), the dynamics of the Loschmidt echo is shown for $L = 12$, 16 and 20 for the parameter values $\mu = V = \omega = 20$ and $\mu = V = \omega = 10$, respectively. For both sets of parameters, we consider three different choices of initial states, $|000111111000\rangle$, $|0000111111110000\rangle$, and $|00000111111111100000\rangle$ for the three different system sizes, with the dynamics of all three states being kinetically constrained to lie within a single Hilbert space fragment consisting of four states, as shown in Fig. 6 for $L = 16$. As a result, all three states execute long-time oscillations in the dynamics, as seen in Figs. 11 (a-b). Furthermore, the envelop of the Loschmidt echo in the first case falls off very slowly compared to the second case. This is due to the larger values of $\mu$, $V$ and $\omega$ in the first case; hence the

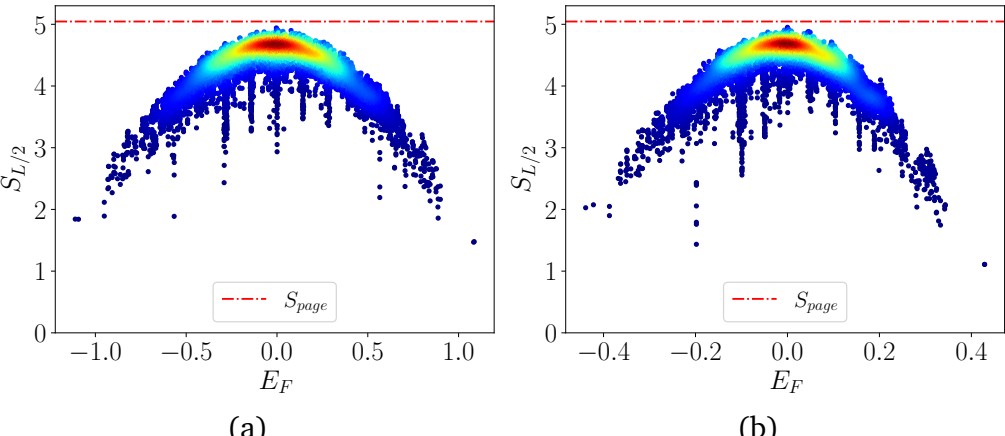

Figure 8: **Entanglement entropy spectrum of the period-2 model for the case of DL and resonances in the intermediate and low driving frequency regime:** (a-b) Plots showing the entanglement entropy $S_{L/2}$ as a function of the Floquet quasienergy $E_F$ obtained from the exact numerical calculation for $J = 1$, $\mu = V = 20$, $L = 16$, and $\omega \approx 6.67$ (third DL point) and $\omega \approx 2.22$ (ninth DL point), respectively. In both cases, we see many low-entanglement states in the middle of the spectrum, which implies that the system breaks ergodicity.

first-order FPT is a better approximation to the exact Floquet Hamiltonian because the higher order corrections are smaller. In Figs. 11 (c-d), we numerically fit the envelop of the Loschmidt echo with time for $L = 20$ for the same parameter values same as in Figs. 11 (a-b). In both cases, we see that the period of oscillations of the Loschmidt echo is almost the same, with $\Delta t \simeq 2\pi$. However, the decay rate of the envelop for the first case is seen to be $1/\tau_1 \simeq 0.0036$, whereas the same quantity for the latter case is almost four times larger, $1/\tau_2 \simeq 0.0139$. The faster decay rate in the second case can be explained by the following argument. Due to the symmetry discussed in Eq. (37), the correction to the first-order FPT effective Hamiltonian will be of third order, which should scale as $J^3/\mu^2$; this is derived in Appendix C at the DL points for $V = 0$. Therefore, the decay rate $1/\tau$, whose dominant contribution is expected to come from the third-order correction, should scale such that $\tau_2/\tau_1 = (\mu_2/\mu_1)^2$. Putting $\mu_1 = 20$ and $\mu_2 = 10$, we expect $1/\tau_2 = 4/\tau_1$, which agrees quite well with the numerically fitted decay rate. In Figs. 12 (a-c), we examine the thermodynamic stability of HSF in a regime with lower frequency by setting $J = 1$, $\mu = V = 20$, $\omega \approx 2.22$ (which corresponds to the ninth DL point), and system sizes $L = 12$, 16 and 18. The rescaled quasienergies for all three cases are again observed to be the same due to the reason mentioned in the earlier case. The low-entanglement states are also found to be quite stable with increasing system size, which indicates the stability of the effective model even in the lower frequency regime. In Fig. 13 (a-b), we show the variation of the half-chain entanglement entropy $S_{L/2}$ with $\mu$ at the stroboscopic number $n = t/T = 2000$ for $\mu = V = \omega$ and $\mu = V = 9\omega$, respectively. We have taken $L = 16$ and the initial state to be $|0000111111110000\rangle$ for this analysis. In both cases, $S_{L/2}(nT)$ gradually decreases with increasing values of $\mu = V$, which shows a slow relaxation behavior in the large driving amplitude regime. It further reveals that there is a smooth crossover from ergodic to non-ergodic behavior occurring in both cases with increasing values of $\mu$ and $V$. In Fig. 13 (a), we observe a crossover occurring for $18 \lesssim \mu = V = \omega \lesssim 30$, which implies a slow relaxation behavior in the high-frequency regime. On the other hand, in Fig. (b), we see a similar behavior occurring for $18 \lesssim \mu = V = 9\omega \lesssim 40$, where $\omega$ lies in the range of $[2, 4.44]$, which necessarily lies in the low to intermediate driving frequency regime. Therefore, we can conclude that DL and resonance-induced HSF in this model is valid

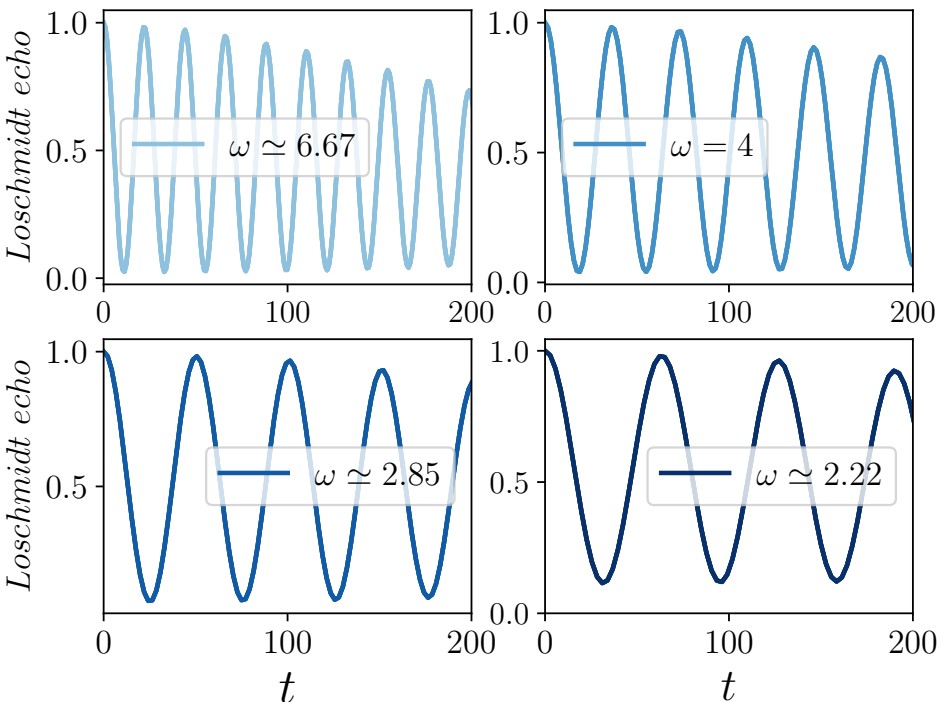

Figure 9: **Dynamics of the Loschmidt echo at other dynamical localization points and at resonance:** Plots of the Loschmidt echo versus time obtained by an exact numerical calculation for $J = 1$, $\mu = V = 20$, $L = 16$, and $\omega \approx 6.67$, $4$, $2.85$, and $2.22$, which correspond to the third, fifth, seventh and ninth DL points, respectively. For all four cases, we consider the initial state to be $|0000111111110000\rangle$, which resides in a HSF cluster consisting of four states as discussed earlier. The Loschmidt echo for all four cases demonstrates long-time revival behavior, indicating that the thermalization is very slow at these parameter values.

for a broad range of driving frequencies even in the thermodynamic limit provided that the amplitude of the driving and the strength of the interaction are strong enough to stabilize this ergodicity-breaking mechanism. It is well-known in the literature that a generic many-body interacting Floquet system is most susceptible to heating in the slow driving regime. However, our model shows that the HSF mechanism emerging from the interplay between DL, interaction and resonances provides significant protection against heating even in the case of slow driving; this is a quite ubiquitous feature of this class of models.

## 5 Period-4 model

We will now discuss a model with the second type of periodic potential, namely, the period-4 model. The general form of the Hamiltonian for this class is given by

$$H = \sum_{j} \left[ J \left( c_j^\dagger c_{j+1} + c_{j+1}^\dagger c_j \right) + \mu(t) \cos(\pi j/2 + \phi) \, c_j^\dagger c_j + V n_j n_{j+1} \right], \tag{46}$$

where $J$ denotes the nearest-neighbor hopping, $\mu$ is the amplitude of the periodic potential with $m = 4$, $V$ defines the nearest-neighbor density-density interaction, and $\phi$ refers to a generalized phase. This model possesses a mirror symmetry [119] for certain special values of $\phi$. Since we are interested in mirror-symmetric configurations for our analysis, it turns out

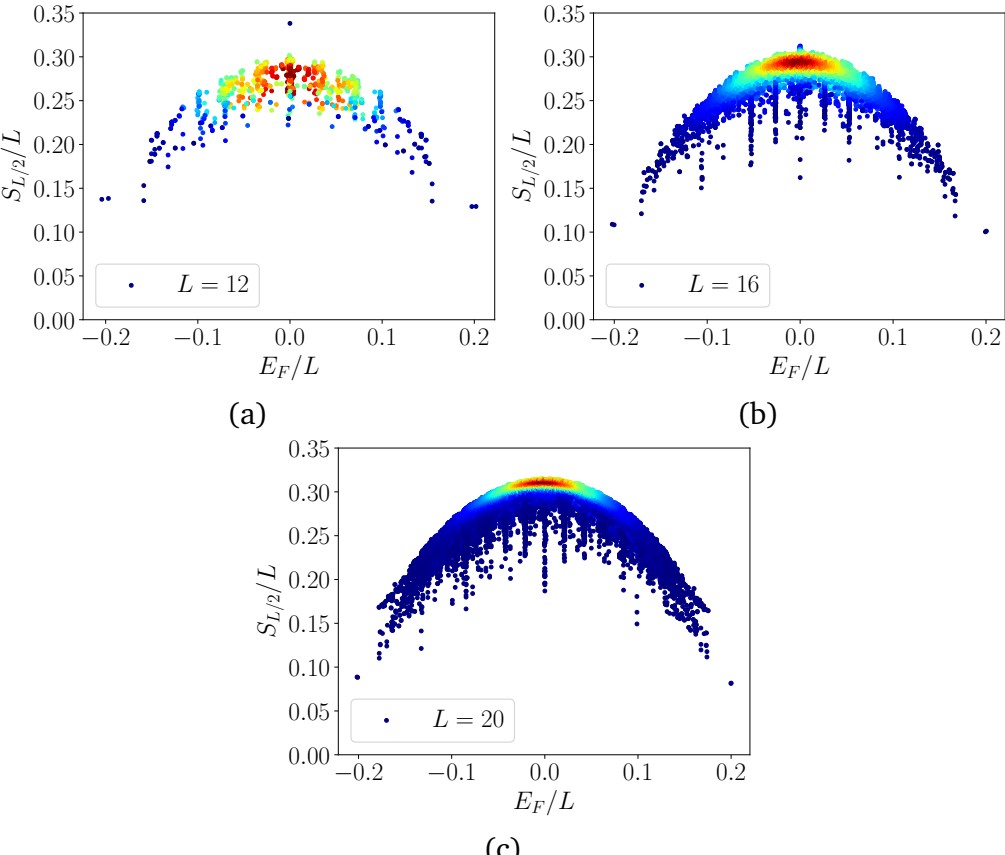

Figure 10: **Scaled entanglement spectrum as a function of scaled quasienergy at another DL point and at resonance for the period-2 model:** (a-c) Plots showing the scaled entanglement entropy $S_{L/2}$ obtained from exact numerical calculations as a function of the scaled quasienergy at a DL point with $J = 1$, $\mu = \omega = V = 20$ for $L = 12$, 16 and 20, respectively. In all three cases, the range of the scaled quasienergy appears to be the same. However, the spectrum of the scaled entanglement entropy broadens with increasing system size, as can be seen in plots (a-c).

that we can only have two possible realizations, namely, $\phi = 0$ and $\phi = 7\pi/4$. We will denote these as type-1 and type-2 cases respectively.

## 5.1 Type-1 mirror-symmetric case

Taking $\mu(t)$ to be proportional to $\mu$ in Eq. (46), we find that the on-site potential pattern for $\phi = 0$ is given by $(\mu, 0, -\mu, 0)$ on four consecutive sites numbered $(4n, 4n+1, 4n+2, 4n+3)$; this is shown in Fig. 14 (a). Assuming $\mu \gg J$, and using the results presented in Appendix B, we find that the first-order FPT Hamiltonian is given by

$$H_{F1}^{(1)} = J \sum_{j=1}^{L/4} \left[ M c_{4j}^\dagger c_{4j+1} + M c_{4j+1}^\dagger c_{4j+2} + M^* c_{4j+2}^\dagger c_{4j+3} + M^* c_{4j+3}^\dagger c_{4j+4} + \text{H.c.} \right],$$

$$M = e^{i\mu T/4} \left( \frac{\sin(\mu T/4)}{\mu T/4} \right). \tag{47}$$

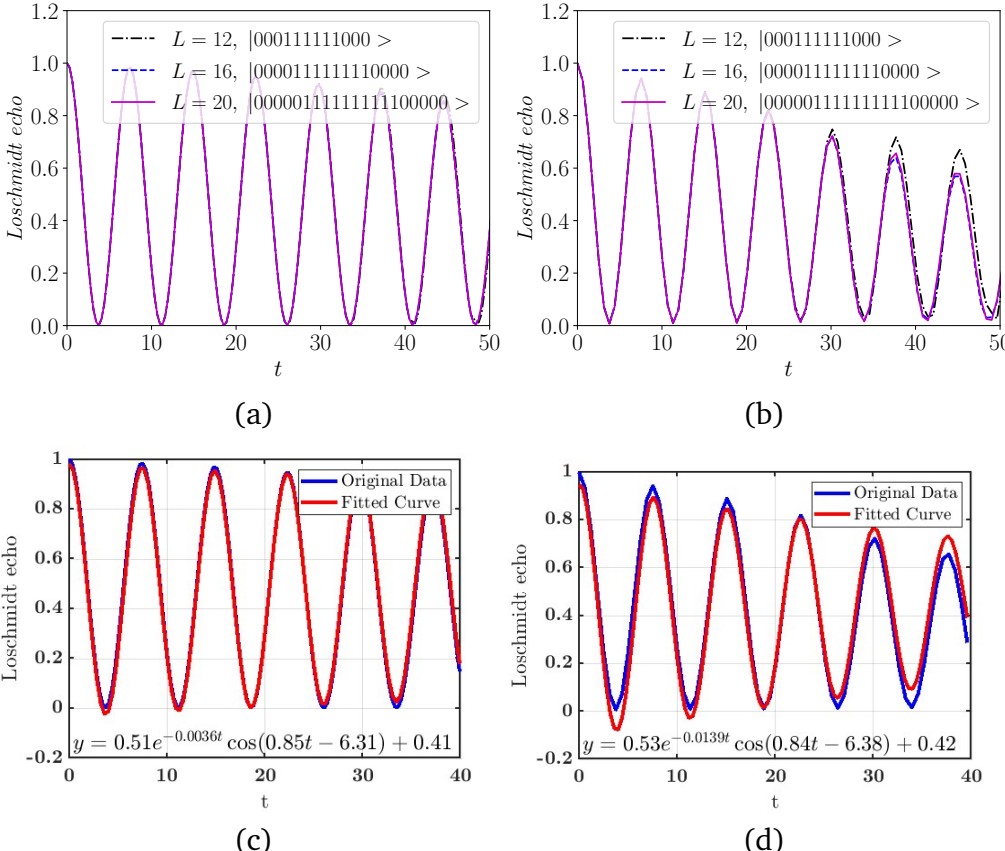

Figure 11: **Variation of Loschmidt echo with time for different system sizes at a DL point and at resonance:** (a-b) Plots showing the variation of Loschmidt echo with time for three different system sizes, $L = 12$, 16 and 20, and $\mu = V = \omega = 20$ and $\mu = V = \omega = 10$, respectively. (c-d) The fitting of the envelop of the Loschmidt echo for $L = 20$ for the same parameter values as in plots (a) and (b). In plots (a-b), we consider three different initial states for the three system sizes, all of them being kinetically constrained to lie within a single Hilbert space cluster due to HSF. Consequently, these states exhibit long-time persistent oscillations. We see that the Loschmidt echo in (a) falls off very slowly compared to (b). Plots (c-d) show the functional forms of the envelops of the Loschmidt echo as extracted from a fitting analysis. In both cases, the period of oscillation of the Loschmidt echo is almost the same, with $\Delta t \simeq 2\pi$. However, the decay rate significantly increases as $\mu$, $V$ and $\omega$ decreases, as is clear from the fitting form of the envelop.

Since the interaction term commutes with the unperturbed Hamiltonian, $H_0$, that part of the Hamiltonian will just be given by

$$H_{F2}^{(1)} = V \sum_{j=1}^{L} n_j n_{j+1}, \tag{48}$$

to first order in $V$.

The symmetry property discussed in Eq. (37) again holds for this model, and therefore, similar to the period-2 case, this system will also exhibit DL when $M = 0$, i.e., when $\mu = 2n\omega$ where $n = 1, 2, 3, \cdots$. Thus, this mirror-symmetric configuration [119] of the period-4 model with $\phi = 0$ and the period-2 model are identical to each other exactly at a DL point. In Figs. 15 (a) and 15 (b), we show the Floquet quasienergy spectrum $E_F$ and the variation of

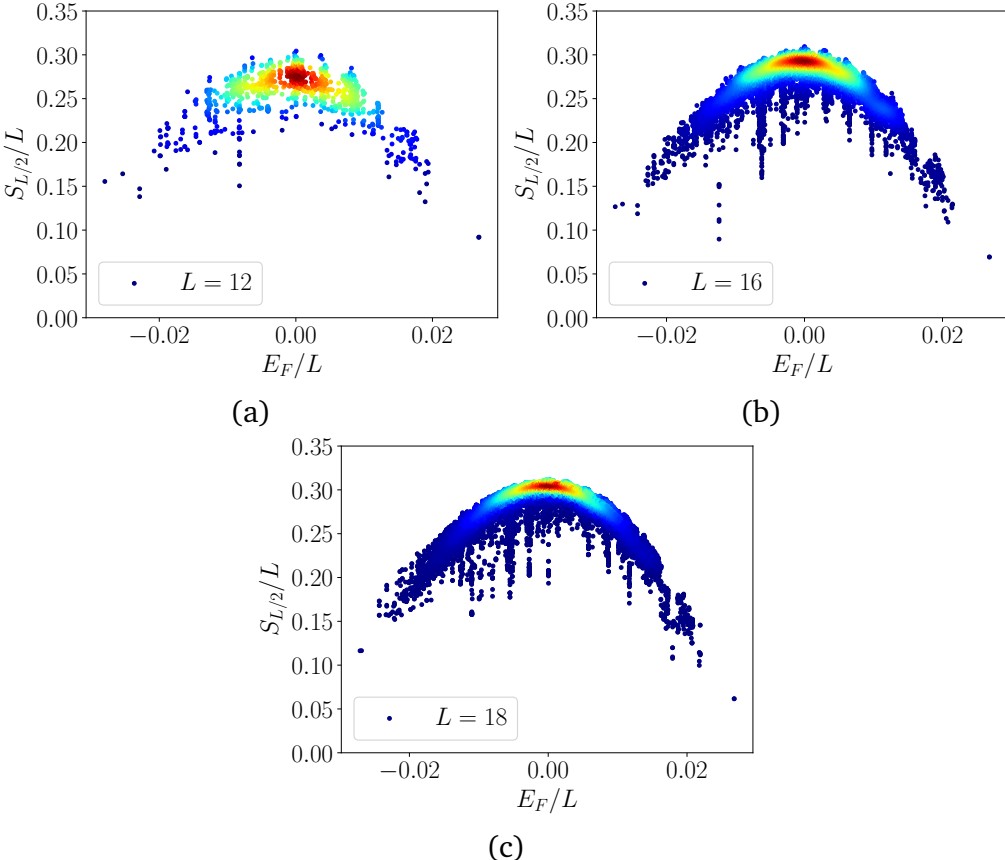

Figure 12: **Scaled entanglement spectrum as a function of scaled quasienergy at another DL point and at resonance for the period-2 model:** (a-c) Plots showing the scaled entanglement entropy $S_{L/2}$ obtained from exact numerical calculations as a function of the scaled quasienergy at a DL point with $J = 1$, and $\mu = V = 20$ and $\omega \approx 2.22$ for $L = 12$, $16$ and $18$, respectively. In all three cases, the range of the scaled quasienergy appears to be the same with many low-entanglement states just as we see in Fig. 10.

the entanglement entropy $S_{L/2}$ with $E_F$ for $J = 1$, $\mu = 20$, $\omega = 20$, and $V = 0.5$. The spectrum looks almost identical to the spectrum of the period-2 model at DL with many low-entanglement states near the middle of the spectrum.

## 5.2 Type-2 mirror-symmetric case

The period-4 model with $\phi = 7\pi/4$ is another mirror-symmetric configuration with many interesting properties, which we will now discuss in detail. Taking $\mu(t)$ to be proportional to $\mu\sqrt{2}$ in Eq. (46), the on-site potential pattern for $\phi = 7\pi/4$ is given by $(\mu, \mu, -\mu, -\mu)$ on four consecutive sites numbered $(4n, 4n+1, 4n+2, 4n+3)$. Assuming $\mu \gg J$, we obtain the following first-order FPT Hamiltonian,

$$H_{F1}^{(1)} = J \sum_{j=1}^{L/4} (c_{4j}^\dagger c_{4j+1} + M_1 c_{4j+1}^\dagger c_{4j+2} + c_{4j+2}^\dagger c_{4j+3} + M_1^* c_{4j+3}^\dagger c_{4j+4} + \text{H.c.}) + V \sum_{j=1}^{L} n_j n_{j+1},$$

$$M_1 = e^{i\mu T/2} \left( \frac{\sin(\mu T/2)}{\mu T/2} \right). \tag{49}$$

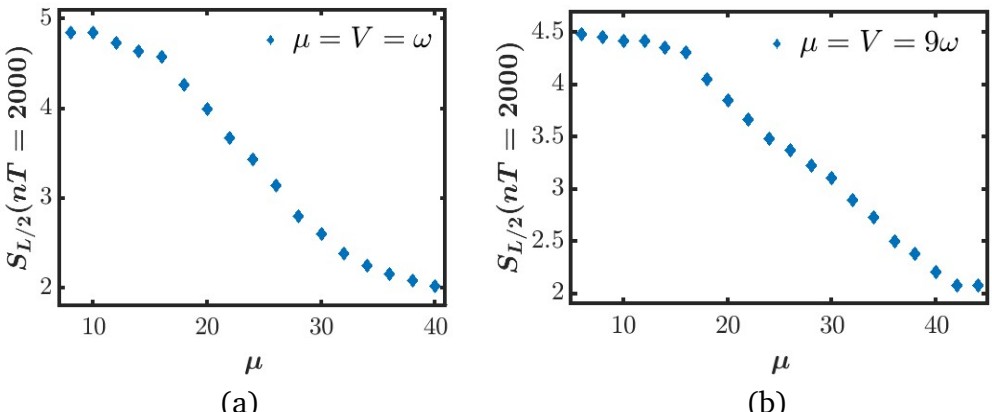

Figure 13: **Slow relaxation behavior in the high and low frequency regimes observed from the dynamics of entanglement entropy at different two dynamical localization and resonance points:** (a-b) The half-chain entanglement entropy $S_{L/2}$ versus $\mu$ at the stroboscopic number $n = t/T = 2000$ for $\mu = V = \omega$ and $\mu = V = 9\omega$, respectively. For this analysis, we have taken $L = 16$ and the initial state to be $|0000111111110000\rangle$. In both cases, we see that $S_{L/2}(nT)$ gradually decreases with increasing values of $\mu = V$, showing a slow relaxation behavior in the large driving amplitude regime. We further see a smooth crossover from ergodic to non-ergodic behavior occurring in both cases with increasing values of $\mu$ and $V$. In Fig. (a), we see the crossover occurring for $18 \lesssim \mu = V = \omega \lesssim 30$, which shows a slow relaxation behavior in the high-frequency regime. In Fig. (b), we observe a similar crossover occurring for $18 \lesssim \mu = V = 9\omega \lesssim 40$, where $\omega$ lies in the range of $[2, 4.44]$. This regime shows a slow relaxation behavior when the ratio of the driving frequency to $J$ lies in the regime of low to intermediate values.

Remarkably, we see that the non-interacting part of the Hamiltonian exactly describes the SSH model [120], with nearest-neighbor hoppings which have alternating strengths given by $J$ and $J|M_1|$. (It is clear that $|M_1|$ is always smaller than 1. The phase of $M_1$ can be removed by doing a unitary transformation of the form $c_j \rightarrow c_j e^{i\alpha_j}$ with appropriately chosen $\alpha_j$'s). We thus see that the periodicity of the model has effectively reduced from 4 to 2.

The expression for $M_1$ implies that this model will exhibit DL for $\mu = n\omega$, where $n = 1, 2, \cdots$ Exactly at these points, the effective first-order Hamiltonian is given by

$$H_{F1}^{(1)} = J \sum_{j=1}^{L/4} (c_{4j}^\dagger c_{4j+1} + c_{4j+2}^\dagger c_{4j+3} + \text{H.c.}) + V \sum_{j} n_j n_{j+1}. \qquad (50)$$

The non-interacting part of this Hamiltonian is an extreme limit of the SSH model with alternating nearest-neighbor hoppings $\gamma_1 = 1$, and $\gamma_2 = 0$. Our model therefore inherits the property of the SSH model that a system with open boundary conditions has topologically protected zero-energy edge modes provided that the hopping strength on the leftmost or rightmost bond is weaker than the strength of the bond next to it. For $\phi = 7\pi/4$, we find that the leftmost bond (between sites numbered 0 and 1) has a hopping strength which is larger than the strength of the next bond, and therefore, the system has no edge modes. However, the system has edge modes for $\phi = \pi/4$, i.e., when the bonds are shifted by one unit cell and the stronger and weaker bonds get interchanged (see the schematic pictures in Fig. 16). In Figs. 17 (a) and 17 (b), we see two zero-energy edge modes and no edge modes for $\phi = \pi/4$ and $\phi = 7\pi/4$, respectively, for $J = 1$, $\mu = \omega = 20$ and $L = 2000$ with open boundary conditions. As shown in Figs. 17 (c) and 17 (d), the two modes are localized at the two edges of

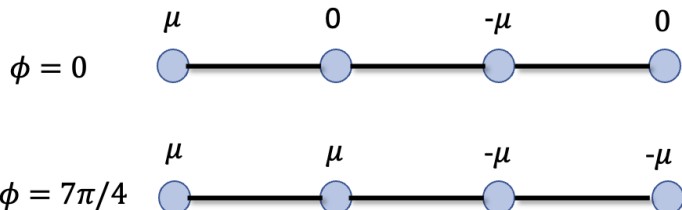

Figure 14: **Schematic of the mirror-symmetric periodic potential pattern for the period-4 model:** Schematic diagrams showing the potential patterns for the two mirror-symmetric configurations of the period-4 model corresponding to $\phi = 0$ and $7\pi/4$.

the system, which can be seen from a plot of the probability $|\psi(j)|^2$ versus the site number $j$. Note that the parameter values $J = 1$ and $\mu = \omega = 20$ imply that the system is at a DL point. Another interesting point to observe is that a static model with a nearest-neighbor hopping $J$ and a period-4 time-independent potential with strength $\mu$ does not have any such zero-energy end modes.

Since the interaction part again commutes with unperturbed Hamiltonian, $H_0$, the effective Hamiltonian to first order in $V$ again reads as $H_{F2}^{(1)} = V \sum n_j n_{j+1}$. The interplay between interaction and DL in this case gives rise to various intriguing phenomena. We will study this in the next few sections using an effective spin model based on the first-order Floquet Hamiltonian.

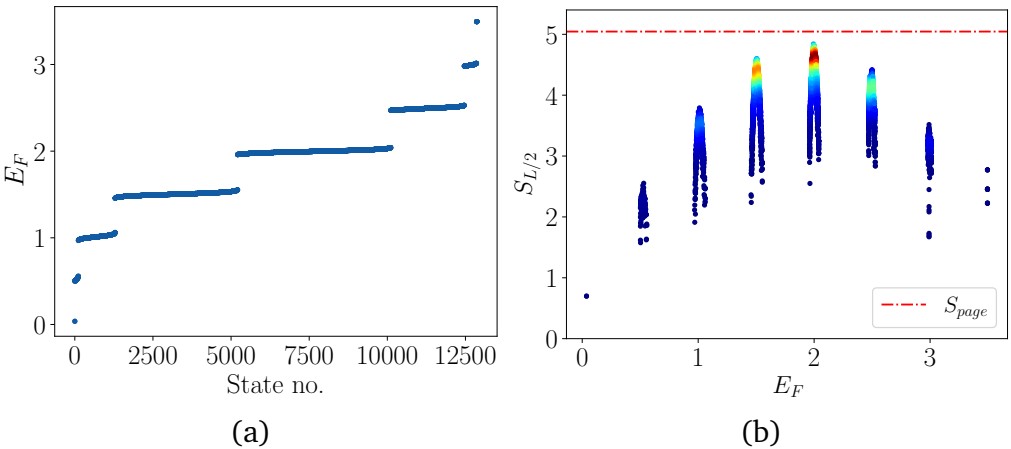

(a)  (b)

Figure 15: **Quasienergy and entanglement entropy spectrum of the period-4 model with $\phi = 0$:** Plots showing the spectrum of (a) $E_F$ and (b) $S_{L/2}$ as a function of $E_F$ at a DL point with $\mu = 2\omega = 20$ and $V = 0.5$. Both the spectra look identical to those of the period-2 case at a DL point.

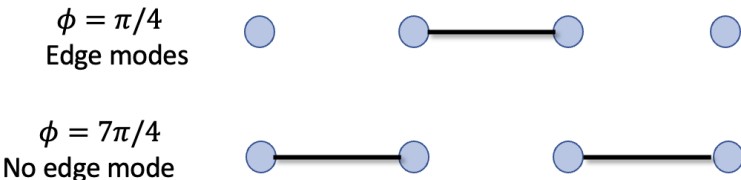

**Figure 16: Schematic of the topologically protected edge modes for the period-4 model at a DL point:** Schematic picture showing topologically protected zero-energy edge modes for the $\phi = \pi/4$ at a DL point and no edge modes for $\phi = 7\pi/4$.

### 5.2.1 Effective spin model based on first-order effective Hamiltonian

To derive the effective spin model, it is convenient to recast the effective Hamiltonian in terms of unit cells

$$H = J \sum_{j=1}^{L/2} \left[ (a_j^\dagger b_j + \text{H.c.}) + V(n_{j,a} n_{j,b} + n_{j,a} n_{j-1,b}) \right], \tag{51}$$

where $a_j$ and $b_j$ denote the annihilation operators on the even and odd numbered sites of the $j$-th unit cell. (Henceforth we will refer to the $j$-th unit cell as the $j$-th site for convenience). The schematic of the periodic-4 model at a DL point is shown in Fig. 18. This above form suggests that the particle number $n_j$ at the $j$-th site (unit cell) commutes with $H_F^{(1)}$. Hence $H_F^{(1)}$ has $L/2$ conserved quantities. (Note that these will be only approximately conserved quantities. The exact effective Hamiltonian will have higher order terms which do not commute with these quantities). For a system consisting of two sites with $n_j^{\max} = 2$, we can have nine possible effective Hamiltonians which are shown in the table below.

Table 4: Allowed configurations and the corresponding effective Hamiltonians for two unit cells at a DL point with $\mu \gg V$ for a period-4 model.

| $n_1$ | $n_2$ | Effective Hamiltonian |
|---|---|---|
| 0 | 0 | $E = 0$ |
| 0 | 2 | $E = V$ |
| 2 | 0 | $E = V$ |
| 2 | 2 | $E = 3V$ |
| 0 | 1 | $H = J(a_2^\dagger b_2 + \text{H.c.})$ |
| 1 | 0 | $H = J(a_1^\dagger b_1 + \text{H.c.})$ |
| 1 | 2 | $H = J(a_1^\dagger b_1 + \text{H.c.}) + V b_1^\dagger b_1 + V$ |
| 2 | 1 | $H = J(a_2^\dagger b_2 + \text{H.c.}) + V a_2^\dagger a_2 + V$ |
| 1 | 1 | $H = J(a_2^\dagger b_2 + a_1^\dagger b_1 + \text{H.c.}) + V n_{b,1} n_{a,2}$ |

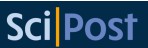

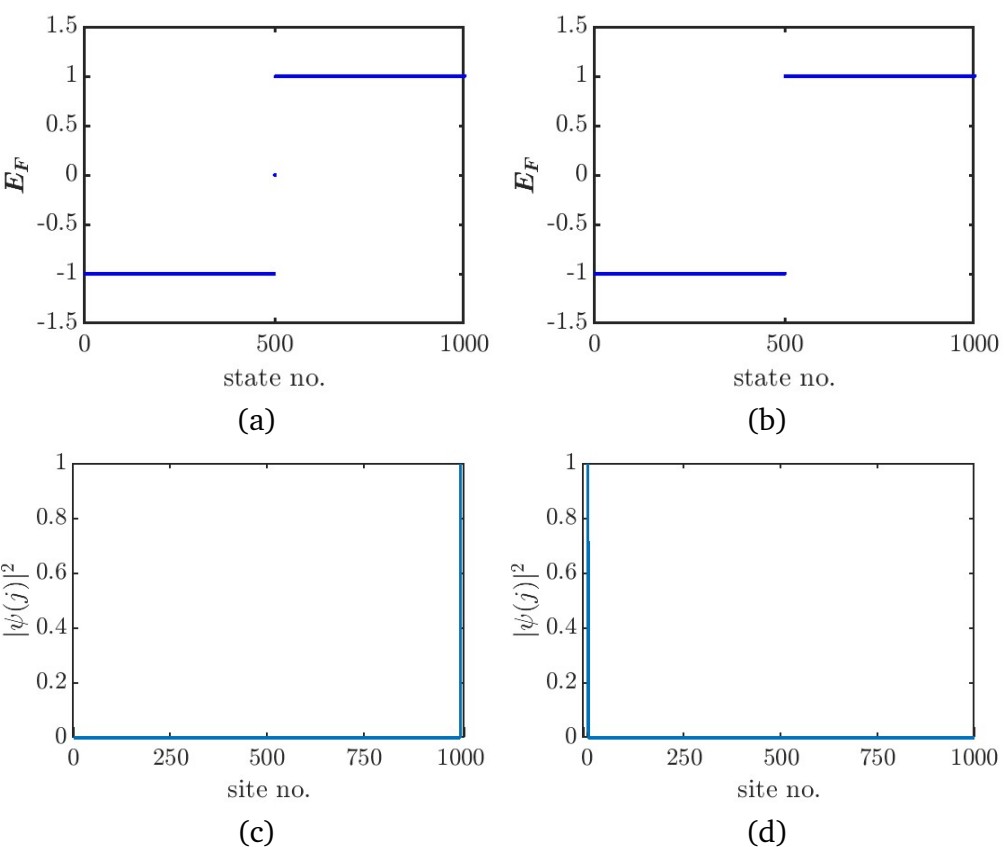

Figure 17: **Quasienergies and wave function probabilities of the edge modes:**
(a-b): Plots of $E_F$ in increasing order versus state number for a system with open
boundary conditions with $J = 1$, $\mu = \omega = 20$. (a) A system with $\phi = \pi/4$ supports
two zero-energy edge modes, while (b) a system with $\phi = 7\pi/4$ does not. (c-d):
Plots showing the probability versus site number for the edge modes showing that
the modes are localized at the ends of the system.

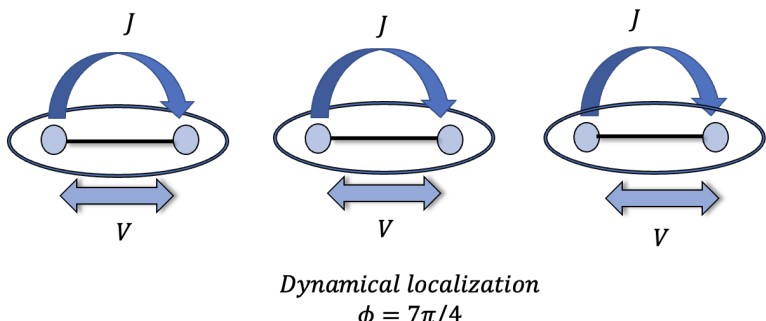

Figure 18: **Schematic of period-4 model with interaction at a DL point:** The
schematic picture of the period-4 model at a DL point, with a nearest-neighbor hop-
ping which alternates between $J$ and zero, and an interaction $V$.

In the table above, $n_1$ and $n_2$ are the occupation numbers of the first and second unit cells, respectively. We can see that eight out of the nine possibilities shown above can be mapped to a non-interacting problem. The only instance when the effects of the interaction is non-trivial is when both the unit cells are singly occupied. For this case, an effective spin degrees of freedom can be defined as

$$|\uparrow\rangle = |10\rangle, \quad |\downarrow\rangle = |01\rangle, \tag{52}$$

where $|10\rangle$ defines a unit cell with the left and the right sites being occupied and empty, respectively, and $|01\rangle$ means the other way around. With this definition, the correlated two-site problem takes the following form

$$H = J\,(\sigma_1^x + \sigma_2^x) + \frac{V}{4}\,(1 - \sigma_1^z)\,(1 + \sigma_2^z), \tag{53}$$

where $\sigma^z$ and $\sigma^x$ are Pauli matrices. This two-site problem can now be generalized to larger system sizes. To do so, we first consider the case where all the sites are singly occupied. The effective spin Hamiltonian for this case is given by

$$H = \sum_{j=1}^{L/2} \left[ J\sigma_j^x + \frac{V}{4}\,(1 - \sigma_j^z \sigma_{j+1}^z) \right], \tag{54}$$

which is essentially the transverse field Ising model with the interaction term $-V/4$ and the transverse field $J$. The other four cases for a system with $L/2 - 2$ unit cells being singly occupied and with the two boundary unit cells being either empty or doubly occupied have effective spin Hamiltonians as follows.

In Table 5, $n_L$ and $n_R$ denote the occupation numbers of the leftmost and rightmost unit cells labeled $j = 1$ and $L/2$, respectively. Therefore, we see from Table 5 that the effective spin Hamiltonian has the form of the transverse field Ising model with additional longitudinal magnetic field terms of strength $\pm V/4$ at the boundary sites, depending on the adjoining sites having $n = 0$ or 2. Before proceeding further, we perform the transformation $\sigma_j^x \to \sigma_j^z$, $\sigma_j^z \to -\sigma_j^x$, and $\sigma_j^y$ remains unchanged. The Hamiltonian then takes the form

$$H = J\sum_{j=2}^{L/2-1} \sigma_j^z - \frac{V}{4}\sum_{j=2}^{L/2-2} \sigma_j^x \sigma_{j+1}^x + \frac{V}{4}\,(\pm\sigma_2^x \mp \sigma_{L/2-1}^x) + \frac{V(L-4)}{8} + \left(0, \frac{V}{2}, \frac{V}{2}, V\right), \tag{55}$$

where the last term depends on the four possible boundary conditions. It may appear that the longitudinal field terms at the boundary sites 2 and $L/2 - 1$ would make it difficult to find the energy spectrum analytically for this model. To overcome this problem, we add two more

Table 5: The four possible effective spin Hamiltonians emerging from a system with $L/2 - 2$ unit cells being singly occupied and two boundary unit cells, each of them either completely occupied or completely empty, at a DL point with $\mu \gg V$ for a period-4 model.

| $n_L$ | $n_R$ | Effective spin Hamiltonian |
|---|---|---|
| 0 | 0 | $H = J\sum_{j=2}^{L/2-1} \sigma_j^x - (V/4)\sum_{j=2}^{L/2-2} \sigma_j^z \sigma_{j+1}^z + (V/4)(-\sigma_2^z + \sigma_{L/2-1}^z) + V(L-4)/8$ |
| 0 | 2 | $H = J\sum_{j=2}^{L/2-1} \sigma_j^x - (V/4)\sum_{j=2}^{L/2-2} \sigma_j^z \sigma_{j+1}^z + (V/4)(-\sigma_2^z - \sigma_{L/2-1}^z) + VL/8$ |
| 2 | 0 | $H = J\sum_{j=2}^{L/2-1} \sigma_j^x - (V/4)\sum_{j=2}^{L/2-2} \sigma_j^z \sigma_{j+1}^z + (V/4)(\sigma_2^z + \sigma_{L/2-1}^z) + VL/8$ |
| 2 | 2 | $H = J\sum_{j=2}^{L/2-1} \sigma_j^x - (V/4)\sum_{j=2}^{L/2-2} \sigma_j^z \sigma_{j+1}^z + (V/4)(\sigma_2^z - \sigma_{L/2-1}^z) + V(L+4)/8$ |

sites, labeled 1 and $L/2$, with Pauli operators $\sigma_1^x$ and $\sigma_{L/2}^x$, which couple to $\sigma_2^x$ and $\sigma_{L/2-1}^x$ respectively [121]. The Hamiltonian then becomes

$$H \;=\; J\sum_{j=2}^{L/2-1}\sigma_j^z \;-\; \frac{V}{4}\sum_{j=1}^{L/2-1}\sigma_j^x\sigma_{j+1}^x \;+\; \frac{V(L-4)}{8}\,, \tag{56}$$

where we have ignored some constants. Note that $\sigma_1^x$ and $\sigma_{L/2}^x$ commute with the Hamiltonian. Hence, there are four decoupled sectors of states corresponding to $\sigma_1^x = \pm 1$ and $\sigma_{L/2}^x = \pm 1$. These four sectors precisely cover the four possible combinations of $\pm$ signs in Eq. (55).

The Hamiltonian in Eq. (56) can now be solved analytically by writing it in terms of Majorana fermion operators using the Jordan-Wigner transformation,

$$\sigma_j^x = \left(\prod_{i=1}^{j-1}\sigma_i^z\right)\alpha_j\,,$$
$$\sigma_j^y = \left(\prod_{i=1}^{j-1}\sigma_i^z\right)\beta_j\,, \tag{57}$$

where $\alpha_j$, $\beta_j$ are Majorana operators. In terms of these operators, the Hamiltonian takes the form

$$H = -\,iJ\sum_{j=2}^{L/2-1}\alpha_j\beta_j \;-\; \frac{iV}{4}\sum_{j=1}^{L/2-1}\alpha_{j+1}\beta_j\,. \tag{58}$$

Since this Hamiltonian is quadratic in terms of Majorana operators, it describes a non-interacting system and its spectrum can be found exactly [121].

To examine the effects of DL on the thermalization of the system, we consider the variation of the half-chain entanglement entropy $S_{L/2}$ with the quasienergy $E_F$, which gives a static measure of ergodicity. As shown in Figs. 19 (a) and 19 (b), we consider the system at a DL point with $J = 1$, $\mu = 20$, and $\omega = 20$, and take $V$ to be 0.5 and 2, respectively. In the first case, we observe many finger-like structures [122] in the entanglement spectrum, which are due to the presence of an extensive numbers of approximate conserved quantities arising due to the DL. Furthermore, the DL offers many frozen states with extremely low-entanglement values, i.e., $S_{L/2} = \ln 2 \simeq 0.693$ or $2\ln 2 \simeq 1.386$ near the middle of the spectrum. Some of the frozen states can be found easily from the effective Hamiltonian, such as $|22220000\rangle$, $|20220200\rangle$, $|22202000\rangle$ and their translated partners. Nevertheless, as shown in Fig. 19 (b), these finger-like structures are absent for $V = 2$ due to the disappearance of these approximate conserved quantities with increasing interaction strength. The low-entanglement states near the middle of the spectrum are still present, which again indicates that this system would thermalize very slowly. We can, therefore, conclude that this slow thermalization occurs due to two possible mechanisms:

(i) the existence of extensive numbers of conserved quantities arising due to the DL, which grows exponentially with the system size as $3^{L/2}$ (which grows less rapidly than the Hilbert space dimension which goes as $2^L$) [123]. We emphasize again that these quantities are conserved to a good approximation only for $\mu \gg J, V$.

(ii) the presence of many frozen state configurations, which do not participate in the dynamics at a DL point.

In Fig. 19 (c), we consider a system away from a DL point with $\mu = 10$, $\omega = 20$ and $V = 0.5$, and we see that the low-entanglement states have disappeared, signaling that the system should thermalize quickly.

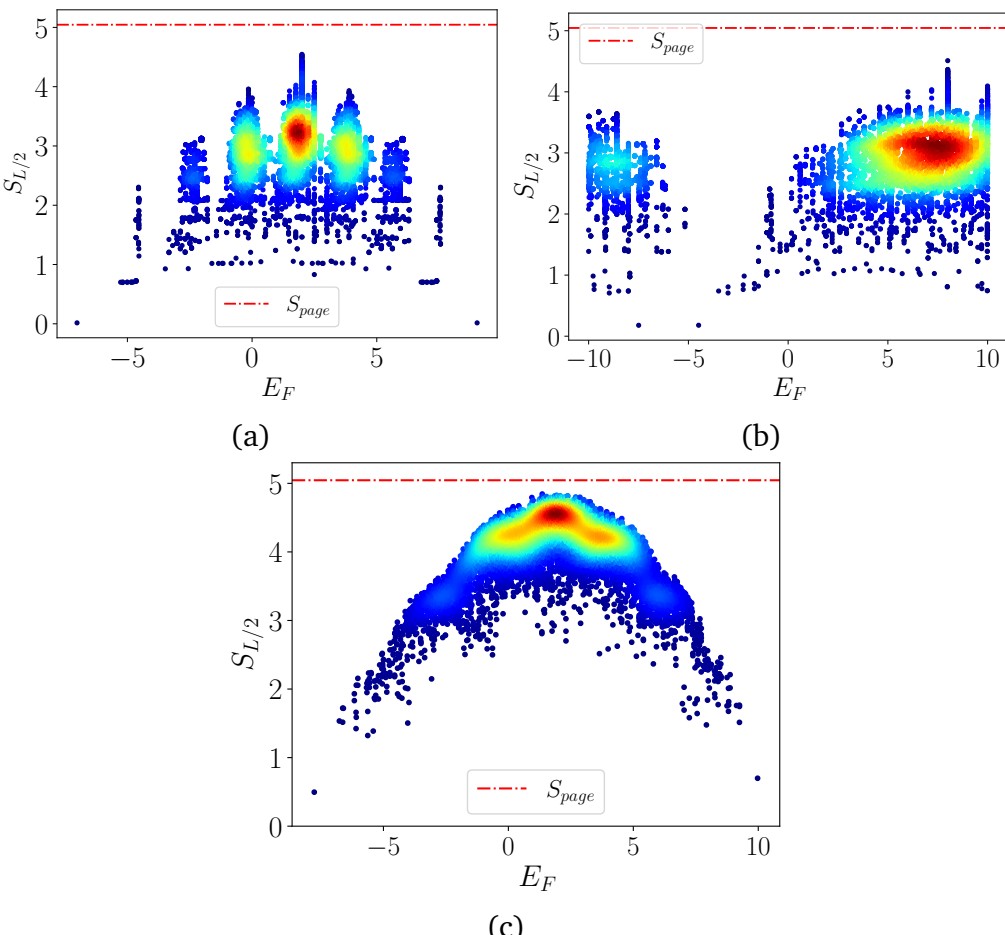

Figure 19: **Entanglement entropy spectrum for the period-4 case at a DL point and away from a DL point:** Plots showing the half-chain entanglement entropy $S_{L/2}$ versus the quasienergy $E_F$. For all three cases, we take $J = 1$ and $\omega = 20$. For the first two cases, we consider a DL point with $\mu = 20$, and (a) $V = 0.5$ and (b) $V = 2$, respectively. (a): The entanglement entropy consists of many finger-like structures with multiple low-entanglement states near the middle of the spectrum. (b): No finger-like structure is present; however, the system still exhibits multiple low-entanglement states near the middle of the spectrum. (c): We consider a point away from DL, with $\mu = 10$, and $V = 0.5$. As opposed to the behavior at a DL point, we see the system exhibits low-entanglement states (except at the end points of the quasienergy spectrum where the entanglement is always low). In all the plots, the color intensity indicates the density of states. In plot (c) we see that the majority of the Floquet eigenstates show thermal entanglement.

To see a dynamical signature of slow thermalization, we study the time evolution of the Loschmidt echo precisely at a DL point with two non-trivial initial states, i.e., $|1\rangle{=}|21012101\rangle$ and $|2\rangle{=}|21102110\rangle$. As shown in Fig. 20 (a), the Loschmidt echo exhibits long-time revivals, showing that the system shows very slow thermalization. To gain an analytical understanding, we first derive an effective Hamiltonian for the state $|210\rangle$, $H_{eff} = \begin{pmatrix} V & J \\ J & 0 \end{pmatrix}$. This Hamiltonian has the energy eigenvalues $E_\pm = \frac{V}{2} \pm \sqrt{J^2 + \frac{V^2}{4}}$. Taking this into account, we see that the initial state $|1\rangle$ would have the highest overlap with $4 \times 2^4 = 64$ such Floquet eigen-

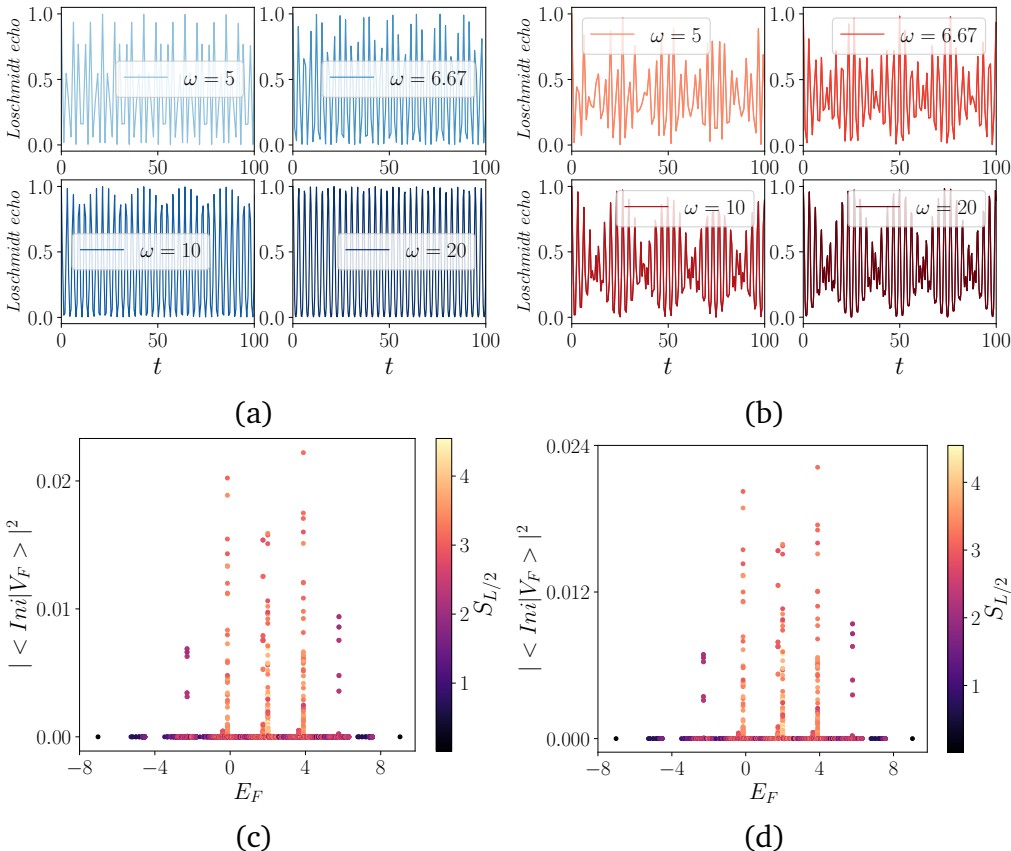

Figure 20: **Dynamics of the Loschmidt echo starting from an initial state and the overlap with Floquet eigenstates at a DL point for the period-4 model:** (a-b): Variation of Loschmidt echo with time for two different initial states ($|Ini\rangle$), $|210120101\rangle$, and $|21102110\rangle$, respectively, for four DL points with $\mu = 20$, $V = 0.5$, and $\omega = 5$, 6.67, 10 and 20 (satisfying $\omega = \mu/n$). In all these cases, the initial states show perfect revivals due to the presence of approximate conserved charges. (c-d): The overlaps of these two initial states with the Floquet eigenstates as a function of $E_F$, with the color bar indicating the variation of $S_{L/2}$. The initial states have significant amounts of overlap with a large number of Floquet eigenstates, and some of these eigenstates have extremely low entanglement.

states with $E_F = 4V$, $4V + 4\sqrt{J^2 + \frac{V^2}{4}}$, $4V - 4\sqrt{J^2 + \frac{V^2}{4}}$, $4V + \sqrt{J^2 + \frac{V^2}{4}}$, $4V + 2\sqrt{J^2 + \frac{V^2}{4}}$, $4V - \sqrt{J^2 + \frac{V^2}{4}}$, and $4V - 2\sqrt{J^2 + \frac{V^2}{4}}$. From the numerically obtained data for a system with $\mu = \omega = 20$, and $V = 0.5$, we find that two Floquet eigenstates with $E_F = 4V = 2$ and $4V + 2\sqrt{J^2 + \frac{V^2}{4}} = 4.06$ have the highest overlaps with this particular initial state, which agrees quite well with the analytically predicted values. Within the approximation of these two highest overlapping states, the Loschmidt echo will have a time-dependence of the form be $|1 + e^{i(E_1 - E_2)t}| = 2|\cos((E_1 - E_2)t/2)|$ which oscillates with a period given by $\Delta t = 2\pi/(E_1 - E_2)$. For the parameter values given above, we find $\Delta t \sim 3$, which agrees with what we see in Fig. 20 (a). In a similar manner, we can find the period of oscillation for the other three DL points with $\omega = 5$, 6.67 and 10. For Fig. 20 (b), we choose the initial state $|2\rangle$ which has the highest overlaps with 64 such Floquet eigenstates. As shown in Fig. 20 (b), we see long-time oscillating behaviors in the Loschmidt echo at the four DL points, which again indicates that the system will not thermalize for a long time. In Figs. 20 (c) and (d), we plot the overlaps of these two initial states, $|1\rangle$ and $|2\rangle$, with all the Floquet eigenstates at a DL point with $\mu = \omega = 20$, where

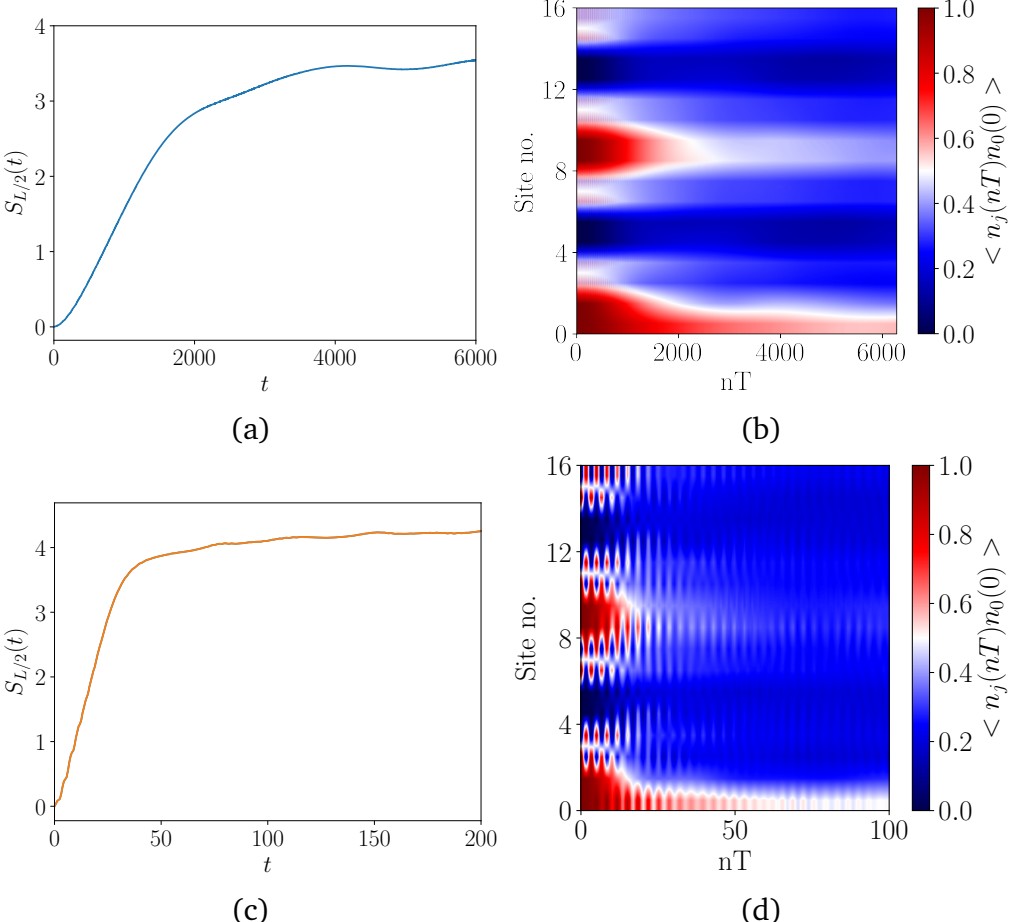

Figure 21: **Dynamics of entanglement entropy and correlation function at a DL point and away from a DL point for the period-4 model:** (a-b): Plots showing the dynamics of $S_{L/2}$ and the correlation function $\langle n_j(nT)n_0(0)\rangle$ for the initial state $|21012101\rangle$ at a DL point with $J = 1$, $\mu = 20$, $\omega = 10$, and $V = 0.5$. (c-d): Same plots away from a DL point with $J = 1$, $\mu = 20$, $\omega = 11$, and $V = 0.5$. (a): At a DL point, we see that the entanglement entropy increases extremely slowly before reaching a saturation value which is less than the thermal value. (b): The correlation function shows a behavior similar to $S_{L/2}$ with a long-time revival pattern. (c): Away from a DL point, $S_{L/2}$ reaches a saturation value soon after an initial growth in time. (d): The correlation function demonstrates a similar behavior, suggesting thermalizing behavior away from a DL point. Note that the time scales in (c-d) are much shorter than in (a-b).

the color bar shows the variation of the entanglement entropy of the Floquet eigenstates. In both cases, we observe that these two initial states have overlaps with multiple Floquet eigenstates, some of them having extremely low entanglement entropy which possibly causes the long- time persistent oscillations in the Loschmidt echo.

To further confirm the above findings, we investigate the dynamics of $S_{L/2}$ and the unequal-time two-point density-density correlation function with the initial state taken to be $|21012101\rangle$. As shown in Figs. 21 (a), we see that the entanglement entropy slowly increases before reaching a saturation value for $\mu = 2\omega = 20$ and $V = 0.5$, as expected. Further, the saturation value ($\sim 3.5$) is much less than $S_{page} \sim 5.1$ signaling a deviation from a volume law behavior. In Fig. 21 (b), we examine the two-point correlation function with the same initial state and see

a behavior similar to $S_{L/2}$. Thus, the dynamics also confirms the behaviors predicted from the static signatures. In Figs. 21 (c) and 21 (d), we repeat the same analysis for the system away from a DL point, namely, for $\mu = 20$, $\omega = 11$, and $V = 0.5$. As opposed to Fig. 21 (a), we see in Fig. 21 (c) that the entanglement entropy saturates quite quickly. In Fig. 21 (d), the two-point correlation function exhibits a similar behavior with the saturation value of $\langle n_j \rangle^2 = 1/4$ at half-filling, which suggests that thermalization has occurred.

Finally, we compare the results obtained from the exact numerics and the first-order FPT calculation for $\mu \gg J$, $V$. In Figs. 22 (a) and 22 (b), we see that the quasienergies obtained from exact numerics and from first-order FPT for $J = 1$, $\mu = \omega = 20$ (lying at a DL point), and $V = 0.5$ agree very well with each other. However, the results for the entanglement entropy do not agree that well. We believe that this disagreement is due to the corrections in FPT which are higher than first-order. The correction terms have a relatively small effect on the Floquet quasienergies but have a noticeable effect on the Floquet eigenfunctions. Consequently, the entanglement entropy deviates significantly from the exact numerical values. Nevertheless, we note that the qualitative behavior of the entanglement entropy is the same in the two cases. In Figs. 22 (c) and 22 (d), we compare the same plots as in Figs. 22 (a) and 22 (b) but for $J = 1$, $\mu = 10$, $\omega = 20$, and $V = 0.25$, which is away from a DL point. Since the first-order term dominates over higher order corrections away from a DL point, we see that the Floquet quasienergy and entanglement entropy agree with each other almost identically.

### 5.2.2 Effects of staggered on-site potential

It is interesting to incorporate the effects of a staggered on-site potential with amplitude $w$ in the period-4 model with $\phi = 7\pi/4$ since such a potential commutes with the unperturbed Hamiltonian $H_0$. Hence, in the presence of such a potential, the first-order FPT Hamiltonian becomes $H_{F,\text{stagg}}^{(1)} = w \sum_{j=1}^{L} (-1)^j n_j$. This term can also be incorporated within the effective spin model. Assuming that all the unit cells are singly occupied (so that there is no boundary field), we obtain

$$H = \sum_{j=1}^{L/2} \left[ \sigma_j^x + \frac{V}{4}(1 - \sigma_j^z \sigma_{j+1}^z) + w\sigma_j^z \right] \tag{59}$$

(In general, the $\sigma^z$ term only appears for unit cells with single occupation, and has no effect on unit cells with $n_j = 0$ or 2). The other sectors where only some of the unit cells have single occupation get modified in a similar way. In Fig. 23, we show the variation of the entanglement entropy $S_{L/2}$ versus $E_F$ for a system with $J = 1$, $\mu = \omega = 20$, $V = 0.5$, and (a) $w = 1$ and (b) $w = 3$. In both cases the spectrum contains multiple finger-like primary structures with further secondary fragments [124], and the secondary fragments become more prominent with increasing value of the staggered potential. In Fig. 24, the dynamics of the Loschmidt echo is shown for a system with $w = 3$, and $\omega = 5$, 6.67, 10 and 20, respectively. For all four DL points, the Loschmidt echo for the initial state $|21012101\rangle$ exhibits an oscillatory behavior for a long period of time, which implies that the system thermalizes very slowly.

### 5.2.3 Effects of resonances

We have so far discussed cases with $V \ll \mu$ where the system shows very slow thermalization at a DL point. In this section, we will address the effects of resonances at two DL points, i.e., $\mu = \omega = V$ and $\mu = 2\omega = V$. For both cases, we note that $\mu$, $V \gg J$, which is the opposite limit to the previously examined cases, and we want to investigate whether this limit can give rise to non-ergodic behavior similar to the period-2 model. To do so, we will first derive an effective Hamiltonian based on the first-order FPT. Similar to the period-2 case, we will first identify the non-trivial processes for a system with four sites. Due to the periodic pattern of

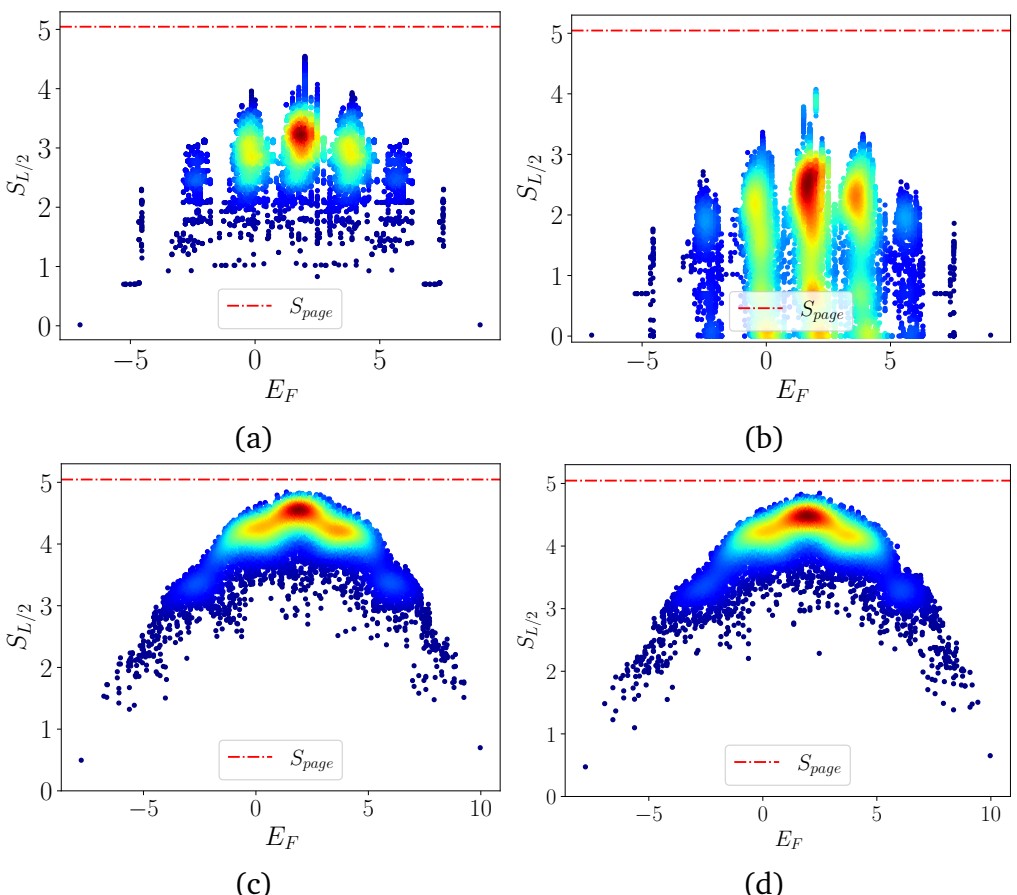

Figure 22: **Comparison of the exact numerical results with the first-order FPT:**
Entanglement entropy $S_{L/2}$ obtained from (a) exact numerical computations and (b)
first-order FPT, for a system with $J = 1$, $\mu = \omega = 20$, and $V = 0.5$ (a DL point). The
quasienergies agree quite well; however, $S_{L/2}$ turns out to be generally smaller from
the first-order FPT in comparison to the exact numerically obtained values. (c-d)
show the same plots as in (a-b) but for a system with $J = 1$, $\mu = 10$, $\omega = 20$, and
$V = 0.5$ (away from a DL point). For these parameter values, both the quasienergies
and the entanglement entropy obtained from the exact numerical computation agree
almost perfectly with the first-order FPT results.

the on-site potential, we need to consider a total of sixteen independent processes to construct
the time-dependent effective Hamiltonian. These are shown in Table 6.

As an example, we will derive the effective time-independent first-order FPT Hamiltonian
for the first process shown in Table 6. In this case, the effective time-dependent Hamiltonian
can be written as

$$H(t) = (\mu(t) + V/2) I + (\mu + V/2) \sigma^z + J \sigma^x,$$
$$H_0 = (\mu + V/2) \sigma^z, \quad H_1 = J \sigma^x, \tag{60}$$

where $H_0$ and $H_1$ are the unperturbed Hamiltonian and the perturbation, respectively, and we
will assume that $\mu, V \gg J$. The instantaneous eigenvalues of $H_0$ are $E_\pm = \pm(\mu(t) + V/2)$. The
eigenfunctions corresponding to $E_k^\pm$ are given by $|+\rangle = \begin{pmatrix} 1 \\ 0 \end{pmatrix}$ and $|-\rangle = \begin{pmatrix} 0 \\ 1 \end{pmatrix}$. These two
eigenvalues again satisfy the condition given in Eq. (9). Therefore, following the usual steps

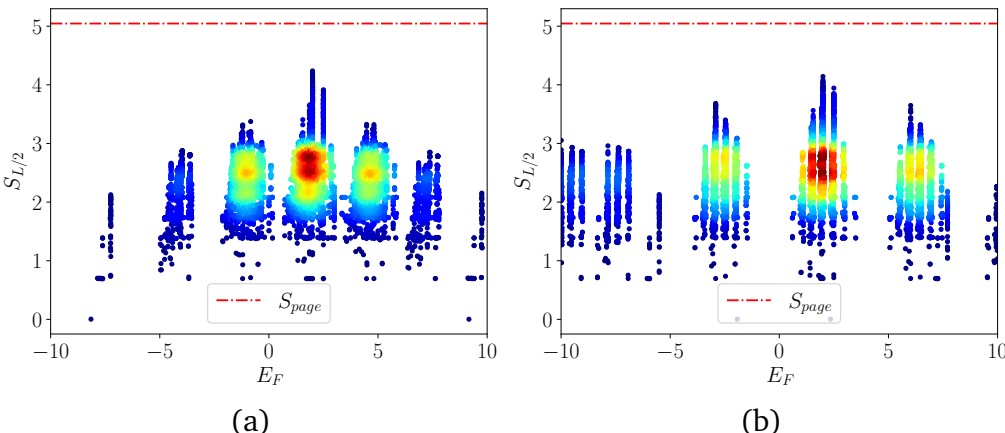

Figure 23: **Entanglement entropy spectrum with a staggered on-site potential at a DL point:** Plots of $S_{L/2}$ versus $E_F$ at a DL point with $J = 1$, $\mu = \omega = 20$, $V = 0.5$, and (a) $w = 1$ and (b) $w = 3$. In both cases we see many finger-like structures with multiple secondary fragments, with the effect becoming clearer with increasing strength of the staggered potential $w$. The color intensity indicates the density of states, revealing that the majority of states demonstrate athermal behavior with $S_{L/2}$ being much smaller than the thermal value.

of degenerate FPT, we obtain

$$\langle +|H_F^{(1)}|+\rangle = 0, \quad \langle -|H_F^{(1)}|-\rangle = 0,$$
$$\langle +|H_F^{(1)}|-\rangle = J\, I(\mu, V, T), \quad \langle -|H_F^{(1)}|+\rangle = J\, I^*(\mu, V, T),$$
$$I(\mu, V, \omega) = \frac{e^{i(2\mu+V)T/4}\sin[(2\mu+V)T/4]}{(2\mu+V)T/2} + \frac{e^{i(2\mu+3V)T/4}\sin[(2\mu-V)T/4]}{(2\mu-V)T/2}. \tag{61}$$

Substituting $\mu = V = \omega$, $I(\mu, V, \omega)$ turns out to be $4i/(3\pi)$. Thus, the effective Hamiltonian for a system consisting of four sites with this specific choice of periodic potential pattern becomes (setting $J = 1$)

$$H_F^{(1)} = \frac{4i}{3\pi}\, n_0\, c_2^\dagger c_1\, (1-n_3) + \text{H.c.} \tag{62}$$

Following the same procedure, we can compute the effective Hamiltonian for all the other processes.

These results enable us to deduce the complete first-order effective Hamiltonian for $\mu = \omega = V$, namely,

$$H = \frac{4i}{3\pi} \sum_{j=1}^{L/2} (-1)^j \left[ (1-n_{2j})c_{2j+2}^\dagger\, c_{2j+1}n_{2j+3} + n_{2j}c_{2j+2}^\dagger c_{2j+1}(1-n_{2j+3}) + \text{H.c.} \right]$$
$$+ \sum_{j=1}^{L/2} \left[ (1-n_{2j+1})\, c_{2j+3}^\dagger c_{2j+2}(1-n_{2j+4}) + n_{2j+1}c_{2j+3}^\dagger c_{2j+2}n_{2j+4} + \text{H.c.} \right]. \tag{63}$$

The form of this Hamiltonian suggests that certain nearest-neighbor hoppings are forbidden, as elaborated below, when the DL and resonance condition are simultaneously satisfied; in principle this can lead to an anomalous thermalization behavior. In Fig. 25, we show the variation of $S_{L/2}$ with $E_F$ obtained from exact numerical calculations and from the first-order FPT analysis, for $J = 1$, and $\mu = \omega = V = 20$. As anticipated, the middle of the entanglement spectrum consists of many low-entanglement states possibly due a fragmented nature of the Hilbert

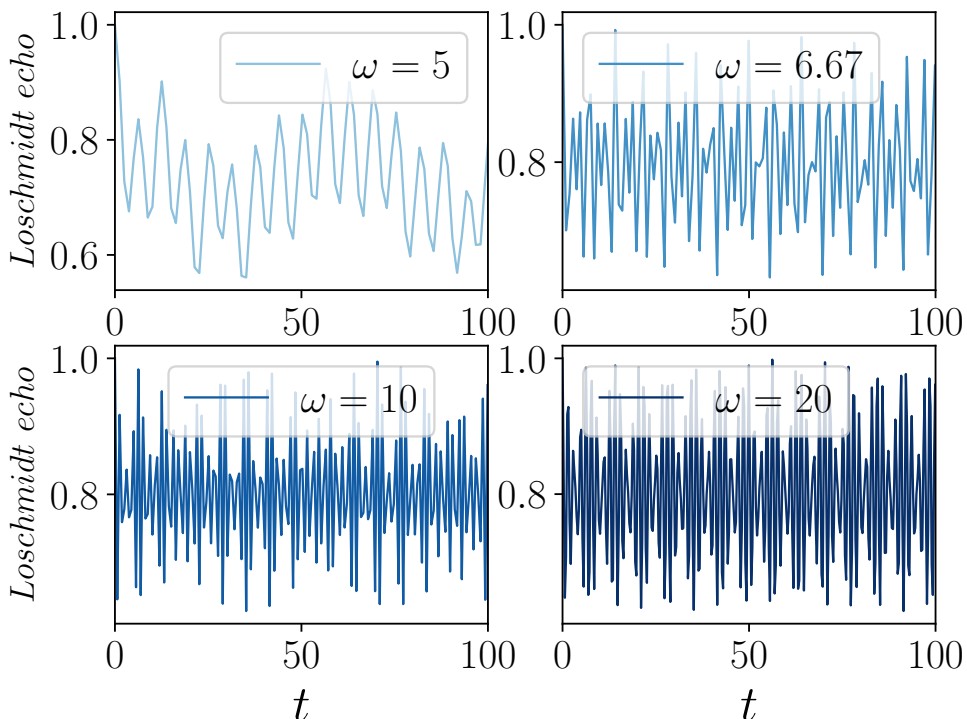

Figure 24: **Dynamics of Loschmidt echo in the presence of a staggered potential:** Plots showing the dynamics of Loschmidt echo for the initial state $|21012101\rangle$ at four DL points with $J = 1$, $V = 0.5$, $w = 3$, $\mu = 20$, and $\omega = 5$, $6.67$, $10$ and $20$. In all four cases, the Loschmidt echos show an oscillatory behavior with an extremely slow decay, suggesting that the system retains the information of the initial state for a long period of time.

space as described below. For example, we can see that this effective Hamiltonian supports a simple fragment which consists of only one state, $|1100110011001100\rangle$, and its translated partners. To construct this single fragment, we note the following constraints following from Eq. (63). The hopping on the bonds $(2j + 1, 2j + 2)$ is only possible if the neighboring sites have $(n_{2j}, n_{2j+3}) = (0, 1)$ or $(1, 0)$. However, the hopping on the bonds $(2j+2, 2j+3)$ requires the neighboring sites to have $(n_{2j+1}, n_{2j+4}) = (0, 0)$ or $(1, 1)$. These two constraints enables us to show that the above state forms a fragment on its own, and it does not mix with other states due to the action of the Hamiltonian. In Fig. 26 (a) and (b), we study the dynamics of the Loschmidt echo for the initial state $|1100110011001100\rangle$ using the exact Floquet dynamics and the first-order effective Hamiltonian, respectively. In both cases, we find that the Loschmidt echo stays very close to 1, with some small oscillations in (a). In Fig. 26 (c), we see that these initial states have highest overlap with two mid-spectrum Floquet eigenstates with $S_{L/2} = \ln 2$. Since $S_{L/2}$ for this initial state is much smaller than $S_{page}(\simeq 5.1$ for $L = 16)$, this state is likely to retain its initial memory for a long period of time.

Now we will consider the case $\mu = 2\omega = V$. Although both $\mu = \omega$ and $\mu = 2\omega$ give rise to DL for a non-interacting system, the presence of $V$ makes a significant difference in the effective Hamiltonian description. Going through the same procedure as before, we obtain the effective Hamiltonian

$$H = \sum_{j=1}^{L/2} \left[ (1 - n_{2j+1})(1 - n_{2j+4}) + n_{2j+1} n_{2j+4} \right] (c_{2j+2}^{\dagger} c_{2j+3} + \text{H.c.}). \tag{64}$$

Table 6: Allowed processes and their corresponding effective time-dependent Hamiltonians for a four-site system with all possible patterns of the periodic potential in a period-4 model in which both resonances and dynamical localization are present.

| Pattern of periodic potential | Process | Effective time-dependent Hamiltonian |
|---|---|---|
| + + - - | $1100 \leftrightarrow 1010$ | $H(t) = \begin{pmatrix} 2\mu(t)+V & J \\ J & 0 \end{pmatrix}$ |
| + + - - | $0100 \leftrightarrow 0010$ | $H(t) = \begin{pmatrix} \mu(t) & J \\ J & -\mu(t) \end{pmatrix}$ |
| + + - - | $0101 \leftrightarrow 0011$ | $H(t) = \begin{pmatrix} 0 & J \\ J & -2\mu(t)+V \end{pmatrix}$ |
| + + - - | $1101 \leftrightarrow 1011$ | $H(t) = \begin{pmatrix} V+\mu(t) & J \\ J & V-\mu(t) \end{pmatrix}$ |
| + - - + | $1100 \leftrightarrow 1010$ | $H(t) = \begin{pmatrix} V & J \\ J & 0 \end{pmatrix}$ |
| + - - + | $0100 \leftrightarrow 0010$ | $H(t) = \begin{pmatrix} -\mu(t) & J \\ J & -\mu(t) \end{pmatrix}$ |
| + - - + | $0101 \leftrightarrow 0011$ | $H(t) = \begin{pmatrix} 0 & J \\ J & V \end{pmatrix}$ |
| + - - + | $1101 \leftrightarrow 1011$ | $H(t) = \begin{pmatrix} \mu(t)+V & J \\ J & V+\mu(t) \end{pmatrix}$ |
| - - + + | $1100 \leftrightarrow 1010$ | $H(t) = \begin{pmatrix} -2\mu(t)+V & J \\ J & 0 \end{pmatrix}$ |
| - - + + | $0100 \leftrightarrow 0010$ | $H(t) = \begin{pmatrix} -\mu(t) & J \\ J & \mu(t) \end{pmatrix}$ |
| - - + + | $0101 \leftrightarrow 0011$ | $H(t) = \begin{pmatrix} 0 & J \\ J & 2\mu(t)+V \end{pmatrix}$ |
| - - + + | $1101 \leftrightarrow 1011$ | $H(t) = \begin{pmatrix} V-\mu(t) & J \\ J & V+\mu(t) \end{pmatrix}$ |
| - + + - | $1100 \leftrightarrow 1010$ | $H(t) = \begin{pmatrix} V & J \\ J & 0 \end{pmatrix}$ |
| - + + - | $0100 \leftrightarrow 0010$ | $H(t) = \begin{pmatrix} \mu(t) & J \\ J & \mu(t) \end{pmatrix}$ |
| - + + - | $0101 \leftrightarrow 0011$ | $H(t) = \begin{pmatrix} 0 & J \\ J & V \end{pmatrix}$ |
| - + + - | $1101 \leftrightarrow 1011$ | $H(t) = \begin{pmatrix} -\mu(t)+V & J \\ J & V-\mu(t) \end{pmatrix}$ |

Note that there is no hopping on the bonds $(n_{2j+1}, n_{2j+2})$. This implies that the occupation number $n_{2j} + n_{2j+1}$ in the $j$-th unit cell commutes with the effective Hamiltonian for all values of $j$. Hence there are $L/2$ approximately conserved quantities, which can protect some of the mid-spectrum states from thermalization for a long time. In Fig. 27, the entanglement entropy spectrum obtained by (a) exact numerics and (b) first-order FPT are shown for $J = 1$, $\mu = 2\omega = V = 20$, and $L = 16$. The fragmentation in the spectrum points towards the existence of conserved charges following from the first-order effective Hamiltonian.

Table 7: First-order effective FPT Hamiltonians corresponding to the allowed correlated processes for a four-site system with all possible patterns of periodic potential in the case of dynamical localization and resonance for a period-4 model.

| Pattern of periodic potential | Process | First-order Floquet Hamiltonian |
|---|---|---|
| $+ + - -$ | $1100 \leftrightarrow 1010$ | $H_F^{(1)} = \frac{4i}{3\pi} n_0 c_2^\dagger c_1 (1 - n_3) + \text{H.c.}$ |
| $+ + - -$ | $0100 \leftrightarrow 0010$ | $H_F^{(1)} = 0$ |
| $+ + - -$ | $0101 \leftrightarrow 0011$ | $H_F^{(1)} = \frac{4i}{3\pi} (1 - n_0) c_2^\dagger c_1 n_3 + \text{H.c.}$ |
| $+ + - -$ | $1101 \leftrightarrow 1011$ | $H_F^{(1)} = 0$ |
| $+ - - +$ | $1100 \leftrightarrow 1010$ | $H_F^{(1)} = 0$ |
| $+ - - +$ | $0100 \leftrightarrow 0010$ | $H_F^{(1)} = (1 - n_0) c_2^\dagger c_1 (1 - n_3) + \text{H.c.}$ |
| $+ - - +$ | $0101 \leftrightarrow 0011$ | $H_F^{(1)} = 0$ |
| $+ - - +$ | $1101 \leftrightarrow 1011$ | $H_F^{(1)} = n_0 c_2^\dagger c_1 n_3 + \text{H.c.}$ |
| $- - + +$ | $1100 \leftrightarrow 1010$ | $H_F^{(1)} = \frac{4i}{3\pi} n_0 c_2^\dagger c_1 (1 - n_3) + \text{H.c.}$ |
| $- - + +$ | $0100 \leftrightarrow 0010$ | $H_F^{(1)} = 0$ |
| $- - + +$ | $0101 \leftrightarrow 0011$ | $H_F^{(1)} = -\frac{4i}{3\pi} (1 - n_0) c_2^\dagger c_1 n_3 + \text{H.c.}$ |
| $- - + +$ | $1101 \leftrightarrow 1011$ | $H_F^{(1)} = 0$ |
| $- + + -$ | $1100 \leftrightarrow 1010$ | $H_F^{(1)} = 0$ |
| $- + + -$ | $0100 \leftrightarrow 0010$ | $H_F^{(1)} = (1 - n_0) c_2^\dagger c_1 (1 - n_3) + \text{H.c.}$ |
| $- + + -$ | $0101 \leftrightarrow 0011$ | $H_F^{(1)} = 0$ |
| $- + + -$ | $1101 \leftrightarrow 1011$ | $H_F^{(1)} = n_0 c_2^\dagger c_1 n_3 + \text{H.c.}$ |

# 6 Experimental accessibility

Flat-band induced quantum many-body scars and HSF have been observed in recent years in the context of equilibrium systems. One of the common mechanisms for these is compact localization which requires special kinds of lattice structures. However, DL-induced flat bands can appear in extremely simple lattice models; therefore, this mechanism is experimentally advantageous compared to the systems demonstrating compact localization. This is one of the main reasons that motivated us to pursue this idea. Moreover, the intricate interplay between DL and resonances in this class of models induces a HSF which is different from the models of HSF discussed in the literature until now. A well-studied model of HSF has a conservation of the total particle number and the total dipole moment [105, 106]. However, the class of models studied here does not conserve the dipole moment but conserves a staggered Ising interaction (see Appendix B); hence this model merits a more detailed investigation. Further, our model for HSF does not seem to arise from any limit of an equilibrium model, unlike an earlier model of HSF which appear in the large $V$ limit of a model of spinless fermions which have a nearest-neighbor interaction with strength $V$ [105, 106]. Therefore, periodic driving is necessary to realize this new kind of non-equilibrium HSF. Furthermore, the kind of driving we chose for our proposed models can be realized in cold-atom systems, and therefore opens up new possibilities for further explorations in experimental settings.

# 7 Discussion

The central results presented in our paper are as follows. We have unraveled an intricate dynamical behavior of a class of disorder-free one-dimensional interacting spinless fermionic models with a periodically driven on-site potential which is also periodic in space. In the

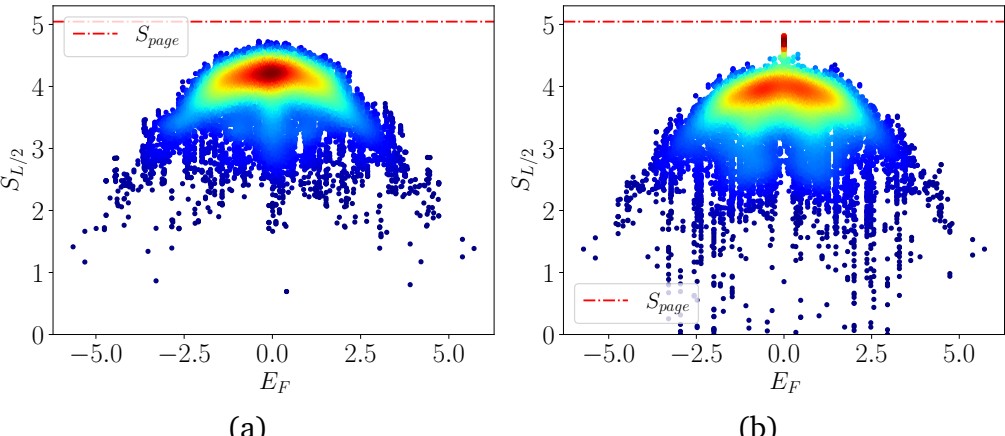

**Figure 25: Entanglement entropy spectrum for a resonant case at a DL point of the period-4 model:** Plots showing the entanglement entropy $S_{L/2}$ versus the quasienergy $E_F$ obtained from (a) an exact Floquet calculation and (b) the first-order FPT Hamiltonian, for a DL point exhibiting a resonance with $J = 1$ and $\mu = \omega = V = 20$. The color intensity indicates the density of states, suggesting that most of the states attain the thermal value. However, there are also many low-entanglement states present near the middle of the spectrum. The quasienergy spectrum obtained from FPT (b) agrees well with the exact numerically computed spectrum (a). However, the entanglement entropy obtained from the first-order FPT is much less than the exact numerically obtained values for many of the states.

absence of interactions, this class of models exhibits DL for a particular set of parameter values, giving rise to one or more flat bands. We have focused in detail on two models, corresponding to potentials with period-2 and period-4 on the lattice. For the period-2 model, we describe a dynamical phase transition which can be observed in the relaxation behavior of correlators in the absence of any interactions. Our investigation shows that a crossover behavior between different power laws of the decay of correlations generally occurs away from the DL points. We find that in the period-2 model, the flat bands which arise due to DL are stable in the presence of a comparatively weak interaction strength due to an emergent integrability. Further, the spectrum of half-chain entanglement entropy as a function of the Floquet quasienergy for a weakly interacting system reveals that there are many low-entanglement states near the middle of the quasienergy spectrum, implying that the system may evade thermalization for a long time. The persistent oscillations in the correlation functions and in the Loschmidt echo which survive for a long period of time support the above statement about thermalization. However, these oscillations decay rapidly in time when we move away from these fine-tuned parameter values.

Remarkably, our model also exhibits Hilbert space fragmentation due to the presence of kinetic constraints when the DL and resonance condition are simultaneously satisfied and the interaction is strong. In the case of period-4, the behavior appears to be much more intriguing, even in the regime of weak interaction. The period-4 model possesses two mirror-symmetric configurations, corresponding to two values of the phase, $\phi = 0$ and $\phi = 7\pi/4$. Our study reveals that the $\phi = 0$ case is identical to the period-2 model at the DL, although the conditions for DL for the two cases are slightly different. The $\phi = 7\pi/4$ model is much more rich compared to the earlier models. In the strong driving limit, the first-order Floquet perturbation theory suggests that the $\phi = \pi/4$ model at the DL points reduces to the SSH Hamiltonian with perfect dimerization, with the hopping amplitudes alternating as $\gamma_1 = 0$ and $\gamma_2 = 1$ respectively. Hence the system supports robust zero-energy edge modes, which are topologically

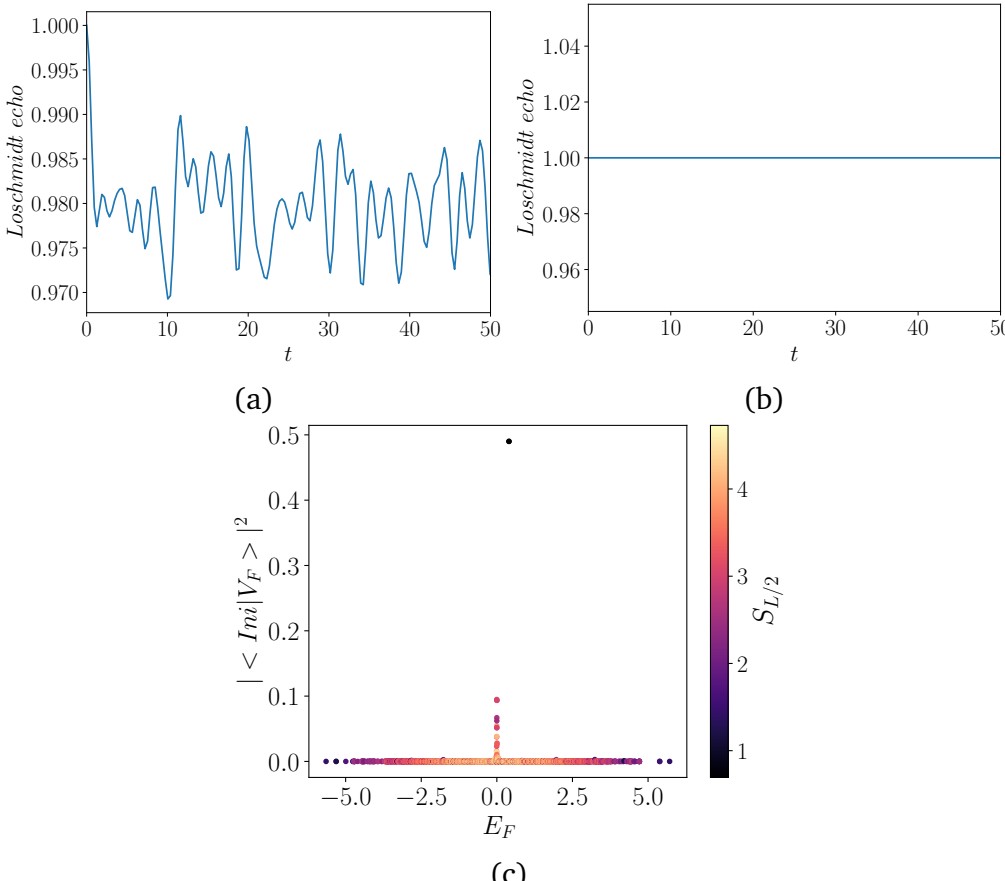

Figure 26: **Dynamics of Loschmidt echo and the overlap with the Floquet eigenstates for a resonant case at a DL point of the period-4 model:** (a) Dynamics of the Loschmidt echo for an initial state ($|Ini\rangle$), $|1100110011001100\rangle$, obtained from (a) an exact numerical calculation and (b) from the first-order effective Hamiltonian, for a system with $J = 1$, $\mu = \omega = 20$, $V = 20$, and $L = 16$. In both cases, the Loschmidt echo is found to stay close to 1, showing that the system retains the memory of the initial state for a long period of time. (c) Overlap of the initial state with the Floquet eigenstates for the same parameter values with the color bar showing the variation of $S_{L/2}$. The Floquet eigenstates having the highest overlap with this initial state lie exactly in the middle of the Floquet eigenvalue spectrum and have extremely low entanglement.

protected. The entanglement spectrum in this regime demonstrates a finger-like structure with many low-entanglement states near the middle of the quasienergy spectrum. We find that there are some initial states which either show long-time persistent oscillations or do not participate in the dynamics at all. We put forward two possible mechanisms for this ergodicity breaking.

(i) There exists an extensive numbers of conserved quantities at the DL points giving rise to sectors which are decoupled from the each other. The number of sectors grows exponentially with the system size as $3^{L/2}$, which is slower than the growth of the Hilbert space dimension $2^L$. We would like to emphasize that these quantities are only approximately conserved quantities; the conservation becomes more and more exact as the driving amplitude is increased.

(ii) Another possibility is that there are many configurations of frozen states which do not evolve with time. However, these states quickly thermalize as we move away from a DL point.

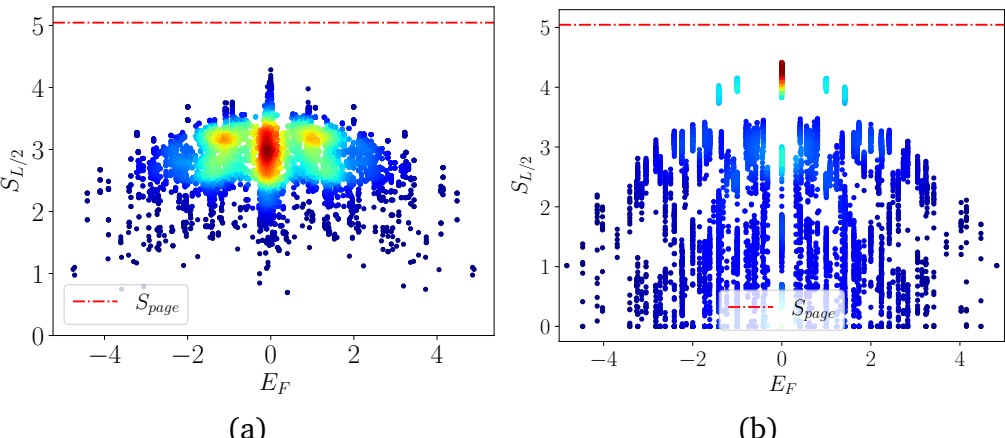

Figure 27: **Entanglement spectrum at another DL point and at resonance for the period-4 model:** Plots showing the entanglement entropy $S_{L/2}$ obtained from (a) exact numerical calculations and (b) first-order Floquet Hamiltonian, at a DL point with $J = 1$, and $\mu = 2\omega = V = 20$. Both calculations show that the quasienergy spectrum consists of multiple fragments with several low-entanglement states lying near the middle of the spectrum. The color intensity indicates the density of states, revealing that the majority of states do not attain the thermal value of te entanglement entropy which is given by the upper envelop of the plots.

In the presence of interactions, the period-4 model with $\phi = 7\pi/4$ is found to have a Floquet Hamiltonian which describes the transverse field Ising model with longitudinal fields at the boundaries. We find it surprising and remarkable that periodic driving of a period-4 model can be tuned to generate well-known systems like the SSH model and the transverse field Ising model which have been extensively studied for many years.

We also discuss the effects of a staggered on-site potential on the DL. In this case, we find that the finger-like structure in the entanglement spectrum further breaks up into secondary fragments, and the Loschmidt echo produces long-time coherent oscillations at the same fine-tuned parameter values. Next, we examine the stability of this non-ergodic behavior whenever the DL and resonance condition are simultaneously satisfied. To study this regime, we choose two different sets of parameter values, $\mu = \omega = V \gg J$, and $\mu = 2\omega = V \gg J$. In both cases, the non-interacting part of the effective Hamiltonian supports DL. For $\mu = \omega = V \gg J$, the entanglement spectrum again demonstrates many low-entanglement states and slow thermalization of the system. In this regime, the effective Floquet Hamiltonian found using first-order perturbation theory shows that DL and resonances together put strict restrictions on the allowed hopping processes, and these restrictions protect some of the mid-spectrum states from thermalization. These kinetic constraints on the dynamics generate dynamically disconnected sectors, a phenomenon called Hilbert space fragmentation. For $\mu = 2\omega = V \gg J$, we see a fractured entanglement spectrum with many segments and with many low-entanglement states near the middle of the quasienergy spectrum. The first-order effective Hamiltonian for this case shows that there are an extensive numbers of conserved quantities. Furthermore, as in the previous case, some processes are again strictly forbidden due to the combination of DL and resonance. These two mechanisms can, in principle, lead the system towards non-ergodic behavior.

We would like to emphasize that the results obtained from the first-order Floquet Hamiltonian (such as the appearance of a large number of conserved quantities) agree well with the results from an exact numerical calculation of the Floquet operator only when (i) the driving amplitude and frequency are much larger than all the other parameters of the system, and (ii)

the time scale of observation of correlation functions and Loschmidt echos is not very large. The two sets of results are expected to deviate from each other at very long times because the effects of higher-order terms in the FPT then become important.

In summary, we have presented a number of models in this paper which can be tailored by Floquet engineering to exhibit rich topological and dynamical phase diagrams which have no counterparts in a time-independent (undriven) model.

# Acknowledgments

S. A. thanks Sumilan Banerjee and Nilanjan Roy for useful discussions.

**Funding information**  S. A. thanks MHRD, India for financial support through a PMRF. D. S. thanks SERB, India for funding through Project No. JBR/2020/000043.

# A   Appendix A

In this appendix we will consider the model described in Eq. (42),

$$H = \sum_{j=1}^{L} (n_j - n_{j+3})^2 \left( c_{j+2}^\dagger c_{j+1} + \text{H.c.} \right), \tag{A.1}$$

where we have set the coefficient in front to be equal to 1 for simplicity, and we assume $L$ to be even. This model has three conserved quantities given by the total particle number and the total staggered Ising interactions ($\sigma_i^z \sigma_{i+1}^z$ where $\sigma_i^z = 2n_i - 1$) on odd- and even-numbered bonds,

$$C_1 = \sum_{j=1}^{L} n_j,$$

$$C_2 = \sum_{j=1}^{L/2} (-1)^j (2n_{2j-1} - 1)(2n_{2j} - 1),$$

$$C_3 = \sum_{j=1}^{L/2} (-1)^j (2n_{2j} - 1)(2n_{2j+1} - 1). \tag{A.2}$$

We will use a transfer matrix method to determine the number of zero-energy states which consist of a single state in the number basis for the Hamiltonian given in Eq. (A.1). We see from that Hamiltonian that there cannot be any hopping between sites $j+1$ and $j+2$ if either
(i) the occupation numbers at sites $(n_j, n_{j+3})$ are either $(0,0)$ or $(1,1)$, or
(ii) the occupation numbers at sites $(n_{j+1}, n_{j+2})$ are either $(0,0)$ or $(1,1)$.
Hence, any configuration which satisfies any of the above conditions for *all* values of $j$ must necessarily be a zero-energy state. This leads us to define an $8 \times 8$ transfer matrix $T_1$ whose rows correspond to the eight possible occupation numbers $(n_j, n_{j+1}, n_{j+2})$, i.e., (111), (110), (101), (100), (011), (010), (001) and (000), and the columns correspond in a similar way to the eight possible occupation numbers $(n_{j+1}, n_{j+2}, n_{j+3})$. The conditions given above imply

that $T_1$ must have the form

$$T_1 = \begin{pmatrix} 1 & 1 & 0 & 0 & 0 & 0 & 0 & 0 \\ 0 & 0 & 1 & 0 & 0 & 0 & 0 & 0 \\ 0 & 0 & 0 & 0 & 1 & 0 & 0 & 0 \\ 0 & 0 & 0 & 0 & 0 & 0 & 1 & 1 \\ 1 & 1 & 0 & 0 & 0 & 0 & 0 & 0 \\ 0 & 0 & 0 & 1 & 0 & 0 & 0 & 0 \\ 0 & 0 & 0 & 0 & 0 & 1 & 0 & 0 \\ 0 & 0 & 0 & 0 & 0 & 0 & 1 & 1 \end{pmatrix}. \tag{A.3}$$

We now have to find the eigenvalues of this matrix. We first note that the matrix has two $4 \times 4$ blocks which are not coupled to each other: the blocks consist of the rows and columns numbered (1235) and (4678), and the two blocks have identical eigenvalues. We can therefore look at either of the two blocks and find the four eigenvalues; the eigenvalues of $T_1$ will then be given by these four eigenvalues, each repeated twice. The block corresponding to (1235) takes the form

$$T_1' = \begin{pmatrix} 1 & 1 & 0 & 0 \\ 0 & 0 & 1 & 0 \\ 0 & 0 & 0 & 1 \\ 1 & 1 & 0 & 0 \end{pmatrix}. \tag{A.4}$$

We find that one of the eigenvalues of $T_1'$ is zero. The other three eigenvalues must therefore be solutions of a cubic equation which turns out to be

$$\lambda^3 - \lambda^2 - 1 = 0. \tag{A.5}$$

The solutions of this equation are found to be

$$\lambda = \frac{1}{3} + \frac{2}{3} \cos \left[ \frac{1}{3} \cos^{-1} \left( \frac{29}{2} \right) - \frac{2\pi k}{3} \right], \tag{A.6}$$

where $k$ can take the values 0, 1, 2. We then find that the three eigenvalues are 1.466 and $-0.233 \pm 0.793i$ approximately. The eigenvalues of the original transfer matrix $T_1$ are therefore given by 1.466, $-0.233 \pm 0.793i$ and 0, each repeated twice. Hence, for a system with a large number of sites $L$, the number of zero-energy states grows exponentially as $1.466^L$ (compared to the total number of states which grows as $2^L$).

It is interesting to compare the number of zero-energy states in this model with the number of such states in a different model which also has a kinetic constraint on the nearest-neighbor hoppings [105, 106]. The Hamiltonian of that model is given by

$$H = \sum_{j=1}^{L} \left[ 1 - (n_j - n_{j+3})^2 \right] \left( c_{j+2}^\dagger c_{j+1} + \text{H.c.} \right). \tag{A.7}$$

This model is known to have three conserved quantities given by the total particle number and the total dipole numbers ($n_i n_{i+1}$) on odd- and even-numbered bonds,

$$C_4 = \sum_{j=1}^{L} n_j,$$

$$C_5 = \sum_{j=1}^{L/2} n_{2j-1} \, n_{2j},$$

$$C_6 = \sum_{j=1}^{L/2} n_{2j} \, n_{2j+1}. \tag{A.8}$$

It has been shown recently that this model can appear as the effective Hamiltonian of a periodically driven system for some special values of the driving parameters [110]. In this model, we see that there cannot be any hopping between sites $j + 1$ and $j + 2$ if either
(i) the occupation numbers at sites $(n_j, n_{j+3})$ are either $(0, 1)$ or $(1, 0)$, or
(ii) the occupation numbers at sites $(n_{j+1}, n_{j+2})$ are either $(0, 0)$ or $(1, 1)$.
To find the number of zero-energy states in this model, we again construct an $8 \times 8$ transfer matrix which now takes the form

$$
T_2 = \begin{pmatrix} 1 & 1 & 0 & 0 & 0 & 0 & 0 & 0 \\ 0 & 0 & 0 & 1 & 0 & 0 & 0 & 0 \\ 0 & 0 & 0 & 0 & 0 & 1 & 0 & 0 \\ 0 & 0 & 0 & 0 & 0 & 0 & 1 & 1 \\ 1 & 1 & 0 & 0 & 0 & 0 & 0 & 0 \\ 0 & 0 & 1 & 0 & 0 & 0 & 0 & 0 \\ 0 & 0 & 0 & 0 & 1 & 0 & 0 & 0 \\ 0 & 0 & 0 & 0 & 0 & 0 & 1 & 1 \end{pmatrix} . \tag{A.9}
$$

This matrix has a $6 \times 6$ block consisting of the rows and columns numbered (124578) and a $2 \times 2$ block consisting of the rows and columns numbered (36) which are not coupled to each other. The $6 \times 6$ block is found to have eigenvalues $(1/2)(1 \pm \sqrt{5})$, $(1/2)(1 \pm i\sqrt{3})$, 0 and 0, while the $2 \times 2$ block has eigenvalues $\pm 1$. Hence the number of zero-energy states in this model grows with the system size as $\tau^L$, where $\tau = (1 + \sqrt{5})/2 \simeq 1.618$ is the golden ratio.

# B  Appendix B

In this appendix, we will derive the first-order FPT for a two-site model with nearest-neighbor hopping $J$, and on-site potentials $\mu_1$ and $\mu_2$, which can then be generalized to a system with larger system sizes. We will further assume that the on-site potential is periodically driven in time with a square pulse protocol. Assuming $\mu_1, \mu_2 \gg J$, we can recast the Hamiltonian as

$$
H = H_0 + H_1, \quad \text{where} \quad H_0 = \begin{pmatrix} \mu_1 f(t) & 0 \\ 0 & \mu_2 f(t) \end{pmatrix}, \quad \text{and} \quad H_1 = \begin{pmatrix} 0 & J \\ J & 0 \end{pmatrix}. \tag{B.1}
$$

The eigenfunctions corresponding to $E_{1,2} = \mu_1 f(t)$ and $\mu_2 f(t)$ are given by $|+\rangle = \begin{pmatrix} 1 \\ 0 \end{pmatrix}$ and $|-\rangle = \begin{pmatrix} 0 \\ 1 \end{pmatrix}$ respectively. The instantaneous eigenvalues $E_{1,2}$ satisfy the degeneracy condition in Eq. (9). Following the procedure for degenerate FPT outlined in Eqs. (11-14), we find that

$$
\langle +|H_F^{(1)}|+\rangle = \langle -|H_F^{(1)}|-\rangle = 0, \quad \langle +|H_F^{(1)}|-\rangle = \langle -|H_F^{(1)}|+\rangle^* = J e^{iB} \left( \frac{\sin B}{B} \right), \tag{B.2}
$$

where $B = (V_1 - V_2)T/4$. Hence $H_F^{(1)}$ is given by

$$
H_F^{(1)} = J e^{iB} \left( \frac{\sin B}{B} \right) (c_1^\dagger c_2 + c_2^\dagger c_1). \tag{B.3}
$$

# C  Appendix C

In this section we will derive the third-order effective Hamiltonian using Floquet perturbation theory for the period-2 model for $V = 0$ at a dynamical localization point obtained from the

first-order effective Hamiltonian. Following the usual steps of perturbation theory, we find that the third-order effective Floquet Hamiltonian is

$$\langle m|H_F^{(3)}T|n\rangle = -\langle m|M^{(3)}|n\rangle + \frac{1}{3}\langle m|(M^{(1)})^3|n\rangle,$$

$$\langle m|M^{(3)}|n\rangle = \sum_{p,q} \langle m|H_1|p\rangle\langle p|H_1|q\rangle\langle q|H_1|n\rangle \int_0^T e^{i\int_0^t (E_m(t_1)-E_p(t_1))\,dt_1}\,dt$$

$$\times \int_0^t e^{i\int_0^{t'}(E_p(t_2)-E_q(t_2))\,dt_2}\,dt' \int_0^{t'} e^{i\int_0^{t''}(E_q(t_3)-E_n(t_3))\,dt_3}\,dt'',$$

$$\langle m|(M^{(1)})^3|n\rangle = \sum_{p,q} \langle m|H_1|p\rangle\langle p|H_1|q\rangle\langle q|H_1|n\rangle \int_0^T e^{i\int_0^t (E_m(t_1)-E_p(t_1))\,dt_1}\,dt$$

$$\times \int_0^T e^{i\int_0^{t'}(E_p(t_2)-E_q(t_2))\,dt_2}\,dt' \int_0^T e^{i\int_0^{t''}(E_q(t_3)-E_n(t_3))\,dt_3}\,dt''. \tag{C.1}$$

While writing Eq. (C.1), we use the fact that the second-order term $M^{(2)}$ is zero for our particular model with our choice of driving protocol due to the symmetry discussed in Eq. (19). Since the perturbation part of the Hamiltonian for our model is off-diagonal in the basis, $|m\rangle = |\pm\rangle$, the only non-zero matrix elements for the third-order effective Hamiltonian are

$$\langle +|H_F^{(3)}|-\rangle = -\langle +|M^{(3)}|-\rangle + \frac{1}{3}\langle +|(M^{(1)})^3|-\rangle, \quad \langle -|H_F^{(3)}|+\rangle = \langle +|H_F^{(3)}|-\rangle^*, \tag{C.2}$$

where

$$\langle +|M^{(3)}|-\rangle = \langle +|H_1|-\rangle\langle -|H_1|+\rangle\langle +|H_1|-\rangle \int_0^T e^{2i\int_0^t \mu(t_1)\,dt_1}\,dt$$

$$\times \int_0^t e^{-2i\int_0^{t'}\mu(t_2)\,dt_2}\,dt' \int_0^{t'} e^{2i\int_0^{t''}\mu(t_4)\,dt_4}\,dt'', \tag{C.3}$$

$$\langle +|(M^{(1)})^3|-\rangle = \langle +|H_1|-\rangle\langle -|H_1|+\rangle\langle +|H_1|-\rangle \int_0^T e^{2i\int_0^t \mu(t_1)\,dt_1}\,dt$$

$$\times \int_0^T e^{-2i\int_0^{t'}\mu(t_2)\,dt_2}\,dt' \int_0^T e^{2i\int_0^{t''}\mu(t_4)\,dt_4}\,dt''. \tag{C.4}$$

For our model, we find using Eq. (8) that

$$\langle +|H_1|-\rangle\langle -|H_1|+\rangle\langle +|H_1|-\rangle = 8\,e^{-ik}\cos^3 k\,. \tag{C.5}$$

For our choice of driving protocol, the integral in Eq. (C.3) can be written as

$$\int_0^T e^{2i\int_0^t \mu(t_1)\,dt_1}\,dt \int_0^t e^{-2i\int_0^{t'}\mu(t_2)\,dt_2}\,dt' \int_0^{t'} e^{2i\int_0^{t''}\mu(t_4)\,dt_4}\,dt''$$

$$= \int_0^{T/2} e^{2i\mu t}\,dt \int_0^t e^{-2i\mu t'}\,dt' \int_0^{t'} e^{2i\mu t''}\,dt''$$

$$+ \int_{T/2}^T e^{-2i\mu(t-T)}\,dt \int_0^t e^{-2i\int_0^{t'}\mu(t_1)\,dt_1}\,dt' \int_0^{t'} e^{2i\int_0^{t''}\mu(t_3)\,dt_3}\,dt''. \tag{C.6}$$

At the dynamical localization points $\mu = n\omega$ obtained from the first-order FPT Hamiltonian, the first integral in Eq. (C.6) is given by

$$\int_0^{T/2} e^{2i\mu t}\, dt \int_0^t e^{-2i\mu t'}\, dt' \int_0^{t'} e^{2i\mu t''} dt'' \;=\; -\frac{T}{4\mu^2}, \quad \text{when} \quad \mu = n\omega. \qquad \text{(C.7)}$$

The second integral in Eq. (C.6) can be written as a sum of three integrals,

$$\int_{T/2}^T e^{-2i\mu(t-T)}\, dt \int_0^t e^{-2i\int_0^{t'} \mu(t_1)\, dt_1}\, dt' \int_0^{t'} e^{2i\int_0^{t''} \mu(t_3)\, dt_3}\, dt''$$

$$= \int_{T/2}^T e^{-2i\mu(t-T)}\, dt \int_0^{T/2} e^{-2i\mu t'}\, dt' \int_0^{t'} e^{2i\mu t''}\, dt''$$

$$+ \int_{T/2}^T e^{-2i\mu(t-T)}\, dt \int_{T/2}^t e^{2i\mu(t'-T)}\, dt' \left( \int_0^{T/2} e^{2i\mu t''}\, dt'' + \int_{T/2}^{t'} e^{-2i\mu(t''-T)}\, dt'' \right). \qquad \text{(C.8)}$$

The three integrals in Eq. (C.8) at the dynamical localization points reduce to

$$\textbf{1.} \quad \int_{T/2}^T e^{-2i\mu(t-T)}\, dt \int_0^{T/2} e^{-2i\mu t'}\, dt' \int_0^{t'} e^{2i\mu t''}\, dt'' \;=\; 0,$$

$$\textbf{2.} \quad \int_{T/2}^T e^{-2i\mu(t-T)}\, dt \int_{T/2}^t e^{2i\mu(t'-T)}\, dt' \int_0^{T/2} e^{2i\mu t''}\, dt'' \;=\; 0,$$

$$\textbf{3.} \quad \int_{T/2}^T e^{-2i\mu(t-T)}\, dt \int_{T/2}^t e^{2i\mu(t'-T)}\, dt' \int_{T/2}^{t'} e^{-2i\mu(t''-T)}\, dt'' \;=\; -\frac{T}{4\mu^2}, \qquad \text{(C.9)}$$

when $\mu = n\omega$. Similarly, we can show that the integral related to $(M^{(1)})^3$ in Eq. (C.4) is

$$\int_0^T e^{2i\int_0^t \mu(t_1)\, dt_1}\, dt \int_0^T e^{-2i\int_0^{t'} \mu(t_2)\, dt_2}\, dt' \int_0^T e^{2i\int_0^{t''} \mu(t_4)\, dt_4}\, dt'' \;=\; 0, \quad \text{when} \quad \mu = n\omega. \qquad \text{(C.10)}$$

Therefore, the third-order effective Hamiltonian for $V = 0$ at the dynamical localization points $\mu = n\omega$ is given by

$$H_F^{(3)} \;=\; \frac{4J^3}{\mu^2} \sum_k \cos^3 k \left( e^{-ik} a_k^\dagger b_k + \text{H.c.} \right). \qquad \text{(C.11)}$$

Interestingly, we see that $H_F^{(3)}$ scales as $J^3/\mu^2$ at the dynamical localization points $\mu = n\omega$, and it does not explicitly depend on $\omega$ at these special points.

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
