# Peer review of "Dynamical localization and slow thermalization in a class of disorder-free periodically driven one-dimensional interacting systems"

_SciPost Physics, doi:SciPost Phys. Core 6, 083 (2023)_

## Round 1 · Referee Report · Anonymous (Referee 1) · 2023-7-7

Strengths

  1. The paper is well written and all calculations are clearly explained.
  2. The results are sound and address the timely topic of quantum non-ergodic behaviour - or failure of thermalization - in Floquet driven quantum many body systems.
  3. The presentation to first focus on non-interacting limits and then add interactions on top is very pedagogical.
  4. The authors find a wealth of different phenomena, from dynamical single particle localization, to kinetic constrained induced quantum many body scars as well as Hilbert space fragmentation.
  5. The mappings to other well studied models, e.g. the SSH chain and the transverse field Ising chain are helpful to appreciate the Floquet engineering aspect.

Weaknesses

  1. While the paper has many different results I did not find a clear motivation why to study this special set-up and model, e.g. is it experimentally easier to assess than other related proposals?
  2. The biggest weakness concerns the novelty of the findings. It seems that each phenomenon - dynamical single particle localization, Floquet driving induced kinetic constraints, Hilbert space fragmentation, slow thermalization from weak (Floquet) integrability , ... have been studied before in other works? What is the new addition which goes beyond any of the existing works?
  3. The employed methods are sound but standard and I do not see any methodological advance (which is not a problem if there is new physics or a new experimental proposal, see point 2 above.)

Report

Overall, I found the paper well written addressing timely topics. However, looking at the Acceptance Criteria Expectations I have a hard time to see the breakthrough or groundbreaking discovery. One could argue that a single model can provide a synergetic link via Floquet engineering from dynamical localization of single particle states to many-body kinetic constraints. However, this has been shown in the literature before. The authors need to explain in how far their study goes beyond previous results and what is the novelty.

Requested changes

  1. The different panels in Figure 1 all show essentially the same except for a different energy scale. There is no need for three panels but the change in scale can be explained in the caption.
  2. Fig.4 c upper panel does show long-lived revivals but there is a clear decay. What is the functional form of the decay and how does it depend on the frequency and parameters?

  • validity: high
  • significance: ok
  • originality: ok
  • clarity: high
  • formatting: good
  • grammar: excellent

Author:  Sreemayee Aditya  on 2023-08-11  [id 3897]

(in reply to Report 1 on 2023-07-07)

Please see the attached file for our response and the list of changes.

Attachment:

response_to_ref1.pdf

---

## Round 1 · Referee Report · Anonymous (Referee 2) · 2023-7-17

Strengths

  • Very well written.
  • Comprehensive and systematic analysis of a class of systems with space-time periodic drive.
  • Very nice combination of analytical and numerical work.

Weaknesses

  • Incomplete analysis of the interacting system and the scaling with system size.

Report

The authors consider a class of periodically driven quantum many-body systems of spinless fermions in one dimension. The Hamiltonian consists of a hopping term and nearest neighbor density-type interactions in the presence of a time periodic on-site potential which is also periodic in space with period $m=2$ and $m=4$. The noninteracting models $V=0$ exhibit the phenomenon of dynamical localization, which at special points in parameter space leads to an effective suppression of the hopping in first order Floquet perturbation theory.

The authors argue that the dynamical localization provides a mechanism for Hilbert space fragmentation since a kinetic constraint emerges. This is reflected in the entanglement properties of the Floquet eigenstates, some of which exhibit an entanglement deficit reminiscent of what is known in quantum many-body scar states in other constrained models.

The paper is very well written and takes a systematic approach to this class of models, analyzing them in detail using a combination of Floquet perturbation theory and exact numerical solutions of the models in the high frequency limit.

My main criticism concerns the analysis of the effect of interactions: It seems that the analysis of the interacting models is essentially limited to a single system size ($L=16$) and there is no systematic analysis of the scaling of the results with $L$. This is particularly important in the interacting case, since the many-body bandwidth grows with $L$ and with growing $L$ and at fixed frequency $\omega$, one should expect that the effect of wrapping quasienergies around the unit circle will become important. In particular, this means that in Figures similar to Fig. 5 I would expect that for increasingly large $L$, the dependence on quasienergy should vanish and it would be extremely interesting to see what happens to the putative scar states in this limit.

It is numerically challenging to access larger system sizes, but perhaps this question can be addressed by lowering the driving frequency and considering several smaller sizes to identify the trend when $L$ increases.

In summary, I find this paper very well written and an interesting systematic analysis of this class of models. For a complete picture, a discussion of the thermodynamic limit in the interacting case is necessary, although the results at finite (fixed) size are of course correct and relevant for small size experiments e.g. in ultracold atomic setups.

If a proper discussion of the scaling with $L$ is provided, this paper is suitable for publication in SciPost Physics Core.

Requested changes

1- The set of references cited for dynamical localization at the end of p. 2 [74-79] seems to be a mistake and probably should be [58-64] (cf. top of the page). 2- Can the authors check if the $\sum_n$ in Eq. (13) needs to exclude $m=n$? Presumably the matrix element of $H_1$ vanishes in this case, but it's perhaps better to be explicit. 3- The figures appear to be rasterized in low resolution. To reach publication quality, the figures should be vectorized. To limit file sizes, in some cases only the panel content should be rasterized (e.g. Fig 4a, 4b etc). 4- In several occasions, the limits $\mu \gg J$ and similar are typeset with a double $>$ sign. It's better to replace this by $\gg$. (p. 6, 8, p. 11, 14, 15, etc.) 5- Fig 2 shows the crossover of the correlation function $\delta C_n$ as a function of the number of cycles. Can the authors show the derived result for the crossover $n_c \approx 1/|\epsilon|^2$ in the figure? 6- The authors show in Fig. 5 the entanglement entropy for exact eigenstates of the Floquet operator and for eigenstates of the 1st order FPT Hamiltonian and say that the quasienergies agree "quite well". It would be interesting to show this comparison directly. One way to do this would be to order both spectra by the phase angle (quasienergy) and then plot $E_{\mathrm{exact}}$ vs. $E_{\mathrm{FPT}}$. A straight line would indicate an exact match, and deviations quantify the agreement. 7- Is the Fig. 5b showing the spectrum of Eq. (36) or Eq. (42)? It would be helpful to indicate this in the figure caption. 8- Fig 5a) and to a larger extent 19b) show states at zero quasienergy with an excess entanglement entropy compared to the rest of the spectrum. This is reminiscent of what is seen in the PXP model, where the origin is a large degenerate subspace at zero energy and the numerically obtained eigenvectors are an arbitrary orthonormal basis of this subspace? If so, there is no physical content to these points and it would be good to check if there is such a degeneracy.

  • validity: high
  • significance: high
  • originality: high
  • clarity: top
  • formatting: excellent
  • grammar: perfect

Author:  Sreemayee Aditya  on 2023-08-11  [id 3898]

(in reply to Report 2 on 2023-07-17)

Please see the file attached for our responses and the list of changes.

Attachment:

response_to_ref2.pdf

---

## Round 2 · Referee Report · Anonymous (Referee 1) · 2023-8-17

Report

The authors have improved the overall motivation of their study and convincingly argued their case w.r.t. the novelty of their results. Given the plethora of phenomena in their model which are explained in a step by step pedagogical manner, I can recommend the publication in SciPost Physics.
  • validity: top
  • significance: good
  • originality: good
  • clarity: high
  • formatting: excellent
  • grammar: excellent

Author:  Sreemayee Aditya  on 2023-09-21  [id 3997]

(in reply to Report 1 on 2023-08-17)

We would like to thank the referee for recommending our paper in SciPost Physics.

---

## Round 2 · Referee Report · Anonymous (Referee 2) · 2023-8-25

Strengths

1- Very well written 2- Nice combination of analytical and numerical work

Weaknesses

1- Results potentially only valid in high frequency regime. 2- Insufficient pixel resolution of figures.

Report

The authors have carefully addressed all comments by the referees and significantly improved and extended the paper.
One major claim of the paper is the stability of dynamical localization in the presence of interactions and the authors have added an analysis as a function of system size in Sec. 5. Unfortunately, this analysis was only carried out in the high frequency regime ($\omega=20$) and it remains unclear whether the discussed phenomena are stable in the thermodynamic limit.
It is well known in Floquet many-body systems that for fixed driving frequency, most systems become ergodic in the limit of large system sizes. This limit is, however, not reachable in practice for high frequencies of the drive. From Fig. 22, I would conclude that the results are very far from the thermodynamic limit, because the density of states is still strongly peaked on the unit circle. However, there are some slight indications in the $L=20$ results in Fig 22. c), that at the edge of the quasienergy spectrum an interference between states at high and low quasienergy starts to take place, "bending up" the entanglement curve. I suspect that in the limit of larger sizes (or, much easier to analyze, at lower driving frequency), the density of states will become flat and potentially the system will be ergodic.
Only in this limit, if the system indeed escapes ergodicity as the authors claim, will the results be convincing in all claimed generality.

This being said, the results and discussion stand on their own and are of course valid at the analyzed system sizes and driving frequencies. It would be good to explicitly say that the results apply for fast driving, though.

In summary, I can recommend this paper for publication in SciPost Physics Core. I cannot recommend publication in SciPost Physics, because of the remaining doubts on the generality of the results in the thermodynamic limit as explained above. This concern prevents fulfilling the acceptance criteria of SciPost Physics.

Requested changes

1- This paper can only be published when the rendering quality of the figures is improved in the production stage.

  • validity: high
  • significance: high
  • originality: high
  • clarity: top
  • formatting: good
  • grammar: perfect

Author:  Sreemayee Aditya  on 2023-09-21  [id 3996]

(in reply to Report 2 on 2023-08-25)

Please see the file attached below for the detailed responses.

Attachment:

ref_response_210923.pdf

---

## Round 2 · Author Response

Dear Editor,

We are hereby resubmitting our paper. We have responded in detail to all the comments by
the two referees and we have made appropriate changes in the manuscript. The list of changes
is given below.

We would like this paper to be considered for SciPost Physics only, not for SciPost Physics Core.
We believe that with the extensive changes that we have made in the manuscript and our detailed
responses to each of the referees' comments, the referees will agree that SciPost Physics is the
appropriate journal for this paper.

Sincerely,

Sreemayee Aditya
Diptiman Sen

---

## Round 2 · List of Changes

We have shown the major changes in blue in the revised manuscript.

  1. We have added two sections, namely, the thermodynamic stability of Hilbert space fragmentation in this class of models (Sec. 5) and the experimental accessibility (Sec. 6).

  2. We have added the derivation of the third-order effective Hamiltonian at the dynamical localization point in Appendix C. This is relevant for answering a question asked by the first referee.

  3. We have removed Fig. 1 (b). However we have retained Fig. 1 (c), which is important for seeing the variation of the bandwidth due to the third-order corrections which become larger when the value of mu is decreased.

  4. We have added two figures (Figs. 23 (a) and (b)) showing the variation of the Loschmidt echo with time for the resonant case of the period-2 model at a dynamical localization point for two sets of parameter values. We have numerically fitted the envelop of the Loschmidt echo varying with time to extract the decay rate of the envelop and have discussed how the decay rate is related to the parameter values used.

  5. We have added a figure (Fig. 2 (e)) showing how the crossover scale n_c diverges as one approaches the critical frequency omega_c from the omega > omega_c side. This further confirms our analytically derived result.

  6. We have added a figure (Fig. 5 (c)) showing a plot of E_exact vs E_FPT for the resonant case of the period-2 model at a dynamical localization point. We have fitted the plot to quantify the agreement of the first-order Floquet perturbation theory with the numerically obtained results.

  7. We have mentioned certain symmetries of our effective Floquet Hamiltonian obtained for the period-2 model at resonance and at a dynamical localization point to contrast this kind of Hilbert space fragmentation (HSF) with the models showing HSF which were known earlier.

  8. We have corrected some typos pointed out by the referees.

---

## Round 3 · Referee Report · Anonymous · 2023-10-30

# Referee report on "Dynamical localization and slow thermalization in a class of disorder-free periodically driven one-dimensional interacting systems"

(Dated: October 30, 2023)

In the resubmission of their manuscript, the authors address the criticism concerning the stability of the claimed Hilbert space fragmentation in the thermodynamic limit by considering their model with a spatial period of two at lower frequencies. Their criterion for dynamical localization is $\mu = V = n\omega$ for $n \in 2\mathbb{Z} + 1$.

[Figure][Figure]

FIG. 1. Left: Spectrum of the Floquet operator of the period-2 model at the dynamical localization point with $n = 9$. Right: Spectrum of the Floquet operator of the period-2 model at the dynamical localization point with $n = 51$.

Unfortunately, I do not find the new analysis convincing to conclude that the phenomena are stable in the thermodynamic limit. The problem is, that the presented new data still shows no sign that the Floquet spectrum covers the unit circle and hence the possibility of many-body resonances is not yet explored. It is however clear that this regime must be reached at larger system sizes, since the many-body bandwidth grows with system size.

To understand the evolution of the spectrum with system size at low frequencies, I have plotted the spectrum of the Floquet operator of Eq. (4) at the dynamical localization point at $n = 9$ at half filling and compared it to $n = 51$ in Fig. 1. Even from the small system sizes I show here, it is clear that in both cases, the system is in a finite size, high frequency regime. Increasing the system size (here, probably a factor 10 in $L$ is required) will eventually wrap the spectrum around the circle many times and create the possibilities for resonances, which may interfere with the discussed Hilbert space fragmentation.

This is a common fallacy in finite size numerics with large model parameters ($V, \mu = 20$, both of the order of the many-body bandwidth). In this "strong interaction regime" as the authors call it, it is essentially impossible to draw conclusions about the limit of $L \to \infty$ from small size data.

I therefore conclude that this paper is a comprehensive study of a class of Floquet models which exhibit interesting behavior at finite size, which is relevant for small scale experiments. It does however not provide sufficient evidence that the observed phenomena form a nonequilibrium phase, stable in the thermodynamic limit.

This paper does not fulfil any of the SciPost Physics "Expectations" in the acceptance criteria.

I hence recommend publication in SciPost Physics Core.

---

## Round 3 · Author Response

Dear Editor,

We are hereby resubmitting our paper in Scipost Physics. In response to the second referee’s comment regarding the generality of Hilbert space fragmentation mechanism in the lower frequency regime, we have now further analyzed the model in this limit both analytically and numerically, and we have made appropriate changes in the manuscript. The list of changes is given below.

  1. The pixel resolution of the figures has been improved in the revised manuscript. All the figures are now in pdf format as requested by the Editor.

  2. To understand HSF in the intermediate and lower frequency regimes, we have examined the period-2 model both analytically and numerically by considering the other DL points at the resonance condition. We have included Figs. 8 and 9 in the revised version of the manuscript, where the entanglement spectrum and the dynamics of Loschmidt echo are investigated in the intermediate and lower frequency regimes.

  3. We have added Fig. 12 showing the results for different system sizes to confirm the thermodynamic stability of the low-entanglement states in the lower frequency regime. This implies non-ergodic behavior of the system in the lower frequency regime.

  4. We have added Fig. 13 in the revised manuscript showing two ergodic to non-ergodic crossovers in terms of the half-chain entanglement entropy at the 2000-th stroboscopic number as a function of the driving amplitude occurring in the high-frequency and low to intermediate frequency regimes at the first and ninth dynamical localization points with resonances, respectively.

We believe that with the changes that we have made in the manuscript and our detailed responses to the referee’s comments, the second referee will agree that SciPost Physics is the appropriate journal for this paper.

Sincerely,

Sreemayee Aditya

Diptiman Sen

---

## Round 3 · List of Changes

1. The pixel resolution of the figures has been improved in the revised manuscript. All the figures are now in pdf format as requested by the Editor.

2. To understand HSF in the intermediate and lower frequency regimes, we have examined the period-2 model both analytically and numerically by considering the other DL points at the resonance condition. We have included Figs. 8 and 9 in the revised version of the manuscript, where the entanglement spectrum and the dynamics of Loschmidt echo are investigated in the intermediate and lower frequency regimes.

3. We have added Fig. 12 showing the results for different system sizes to confirm the thermodynamic stability of the low-entanglement states in the lower frequency regime. This implies non-ergodic behavior of the system in the lower frequency regime.

4. We have added Fig. 13 in the revised manuscript showing two ergodic to non-ergodic crossovers in terms of the half-chain entanglement entropy at the 2000-th stroboscopic number as a function of the driving amplitude occurring in the high-frequency and low to intermediate frequency regimes at the first and ninth dynamical localization points with resonances, respectively.

---

## Editorial Decision

published